# Disentangling strictly self-serving mutations from win-win mutations in a mutualistic microbial community

**Samuel Frederick Mock Hart[†], Jose Mario Bello Pineda[†], Chi-Chun Chen, Robin Green, Wenying Shou\***

Division of Basic Sciences, Fred Hutchinson Cancer Research Center, Seattle, United States

**Abstract** Mutualisms can be promoted by pleiotropic win-win mutations which directly benefit self (self-serving) and partner (partner-serving). Intuitively, partner-serving phenotype could be quantified as an individual's benefit supply rate to partners. Here, we demonstrate the inadequacy of this thinking, and propose an alternative. Specifically, we evolved well-mixed mutualistic communities where two engineered yeast strains exchanged essential metabolites lysine and hypoxanthine. Among cells that consumed lysine and released hypoxanthine, a chromosome duplication mutation seemed win-win: it improved cell's affinity for lysine (self-serving), and increased hypoxanthine release rate per cell (partner-serving). However, increased release rate was due to increased cell size accompanied by increased lysine utilization per birth. Consequently, total hypoxanthine release rate per lysine utilization (defined as 'exchange ratio') remained unchanged. Indeed, this mutation did not increase the steady state growth rate of partner, and is thus solely self-serving during long-term growth. By extension, reduced benefit production rate by an individual may not imply cheating.
DOI: https://doi.org/10.7554/eLife.44812.001

**\*For correspondence:**
wenying.shou@gmail.com

[†]These authors contributed equally to this work

## Introduction

Mutualisms, mutually beneficial interactions between species, are widely observed between microbes (*Goldford et al., 2018*; *Morris et al., 2013*; *Morris et al., 2012*; *Seth and Taga, 2014*) and between microbes and their hosts (*Seth and Taga, 2014*). Often, mutualisms involve the release and consumption of essential metabolites such as vitamins and amino acids (*Beliaev et al., 2014*; *Carini et al., 2014*; *Helliwell et al., 2011*; *Jiang et al., 2018*; *Rodionova et al., 2015*; *Zengler and Zaramela, 2018*). Extensive metabolic interactions between microbes have been thought to contribute to the difficulty of culturing microbes in isolation (*Kaeberlein et al., 2002*). Under certain conditions, microbial metabolic exchanges may even promote community growth (*Tasoff et al., 2015*).

In communities, a mutation can exert direct effects on the individual itself as well as on the partner (see *Figure 1A* for definition of "direct effects"). For example, a mutation can increase benefit supply to the partner wtihout affecting self's growth rate (*Figure 1A*, ii). We classify this mutation as "strictly partner-serving". As another example, a mutation can increase self's growth rate without affecting benefit supply to the partner (*Figure 1A*, iii). By growing better, the mutant stimulates partner growth. However, we classify this mutation as "strictly self-serving" because promoting partner growth is mediated indirectly by improved self-growth. Qualitatively, the direct fitness effect of a mutation on self and on partner can be positive, neutral, or negative, giving rise to $3 \times 3 = 9$ types.

Distinguishing mutation types is important for predicting their evolutionary successes. Consider the general case of microbial mutualisms without any partner choice mechanisms. That is, an

**eLife digest** Many organisms – including microbes – have mutually beneficially relationships. Often, the exchanged goods, such as nutrients, are costly to make. But what happens when individuals evolve to help themselves more? Can they also evolve to be more helpful to others?

Hart, Pineda et al. studied a community of two genetically modified yeast strains that had to exchange essential nutrients to survive. One strain overproduced the molecule hypoxanthine, but depended on the second strain to provide the nutrient lysine, and vice versa. The communities were then allowed to evolve. The lysine-requiring strain frequently ended up with a mutation that initially seemed to be win-win: helping self to grow faster and at the same time, releasing more hypoxanthine to the partner. However, closer examination showed that the mutation also made these cells bigger, and bigger cells had to consume more lysine. Consequently, releasing more hypoxanthine was accompanied by consuming more lysine. Since the 'give' to 'take' ratio stayed the same, the partner strain did not benefit more from the mutant than from the ancestor.

This suggests that an individual should not be considered helpful solely based on how much it gives to a partner, but also, on how much it takes. In the case of the mutant yeast strain, it produced 30 percent more nutrients, but also consumed 30 percent more, and was therefore not more helpful to the partner than the ancestor. Similarly, releasing less may not imply cheating. Beneficial interactions are very common in natural communities, such as among microbes living in the mouth cavity and the gut. Therefore, a better understanding of how they benefit from and affect each other may provide scientists with more insight into diseases linked to problems with microbial communities, such as tooth decay, inflammation of the gut, or obesity.
DOI: https://doi.org/10.7554/eLife.44812.002

individual is not capable of discriminating or 'choosing' among spatially-equivalent partners (*Sachs et al., 2004*; *Shou, 2015*). Then, a well-mixed environment will favor mutations with a positive direct fitness effect on self (i.e. selfish, strictly self-serving, and win-win; *Figure 1B*, 'mixed'). This is because in a well-mixed environment, benefits from mutualistic partners are uniformly distributed, and thus how much an individual contributes to mutualistic partners is irrelevant. In contrast, in a spatially-structured environment, mutations exerting a positive direct effect on the mutualistic partner (i.e. win-win, strictly partner-serving, and altruistic) can be favored, while selfish mutations can be disfavored (*Chao and Levin, 1981*; *Doebeli and Knowlton, 1998*; *Hamilton, 1964*; *Harcombe, 2010*; *Momeni et al., 2013b*; *Nowak, 2006*; *Sachs et al., 2004*; *Shou, 2015*) (*Figure 1B*, 'spatial'). This is because in a spatially-structured environment, interactions are localized and repeated between neighbors. If an individual does not aid its mutualistic neighbor, the individual will eventually suffer as its mutualistic neighbor perishes. Win-win mutations are particularly intriguing because they directly promote both sides of a mutualism.

Here, we analyze a mutation arising during the evolution of an engineered yeast mutualistic community in a well-mixed environment. This community CoSMO (Cooperation that is Synthetic and Mutually Obligatory) (*Shou et al., 2007*) consists of two *S. cerevisiae* strains that interact via metabolite cross-feeding (*Figure 1C*). $L^-H^+$ requires lysine ($L$) and overproduces and releases the adenine derivative hypoxanthine ($H$) (*Hart et al., 2019a*). The complementary $H^-L^+$ requires $H$, and overproduces and releases $L$. The two yeast strains are reproductively isolated, and thus can be regarded as two species. This mutualism is 'cooperative' in the sense that metabolite over-production is costly to both strains (*Figure 1—figure supplement 1*; *Waite and Shou, 2012*). Mutualisms modeled by CoSMO are widely observed in natural communities (*Beliaev et al., 2014*; *Carini et al., 2014*; *Helliwell et al., 2011*; *Jiang et al., 2018*; *Rodionova et al., 2015*; *Zengler and Zaramela, 2018*), including those in the gut and the oral microbiota (*Palmer et al., 2001*; *Rakoff-Nahoum et al., 2014*). A simplified community allows us to gain mechanistic insights into basic questions in microbial ecology and evolution (*Momeni et al., 2011*.) Indeed, principles learned from CoSMO, including how fitness effects of species interactions affect the composition and spatial patterning of member species, and mechanisms that protect mutualisms from exploiters, have been found to operate in communities of non-engineered microbes (references in *Momeni et al., 2013a*; *Momeni et al., 2013b*; *Waite and Shou, 2012*). CoSMO offers an ideal system for examining the evolution of

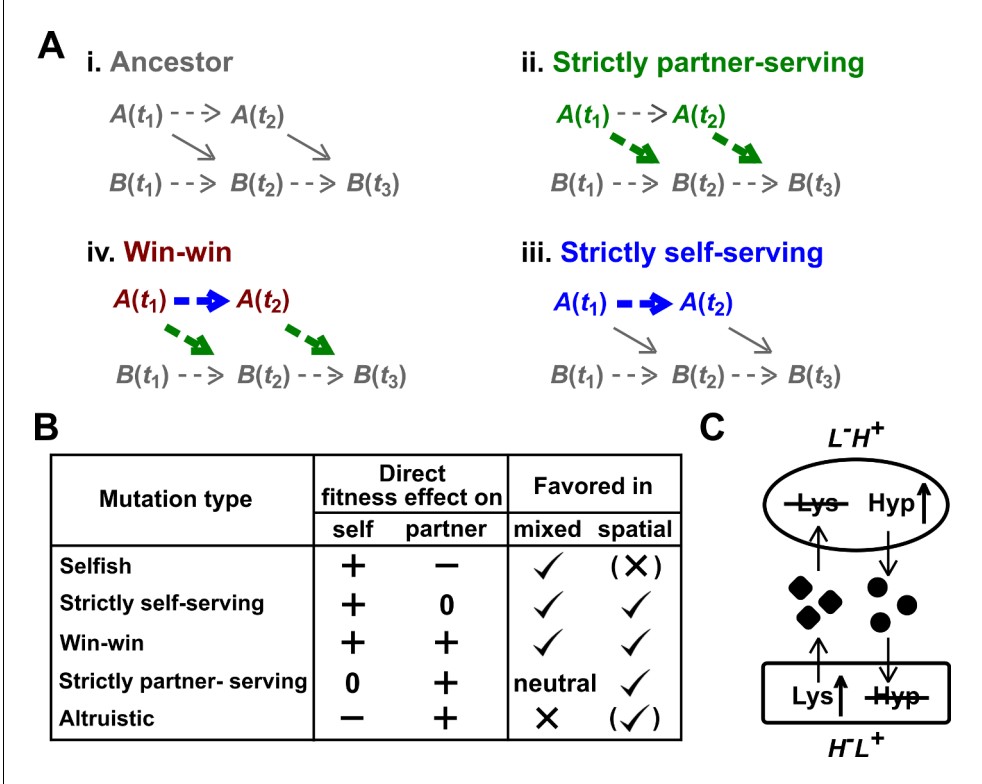

**Figure 1.** Mutation types predicted to be favored during the evolution of a mutualistic community. (**A**) Direct fitness effects. To define direct effects, we use a framework similar to Chapter 10 of *Peters et al. (2017)*. For simplicity, consider a commensal community where population *A* benefits population *B*. (**i**) Populations *A* and *B* grow over time $t_1$, $t_2$, etc. Basal growth rates of monoculture populations are marked by dashed arrows. *A* also releases a metabolite which promotes the growth of *B* (solid diagonal arrow). Thus, $B(t_2)$ will depend on both $B(t_1)$ and $A(t_1)$. (**ii**) Suppose that at time $t_1$, *A* acquires a mutation (green) which does not alter basal growth rate, but increases benefit supply to partner (thick green arrow). This will increase $B(t_2)$ even if we had held the dynamics of mutant *A* to that of the ancestral *A*. We define such a mutation as "strictly partner-serving". (**iii**) At time $t_1$, *A* acquires a mutation (blue) that increases *A*'s basal growth rate (thicker blue arrow), but not benefit supply rate. This mutation will promote $B(t_3)$ via increasing $A(t_2)$, but will not promote $B(t_3)$ if we had held the dynamics of mutant *A* to that of the ancestral *A*. Because increased $B(t_3)$ is indirect (mediated by increased $A(t_2)$), we define this mutation as "strictly self-serving". (**iv**) A win-win mutation (brown). (**B**) Mutation types in a mutualistic community and their evolutionary fates. Mutations that exert a positive direct effect on self (selfish, strictly self-serving, and win-win) are favored in a well-mixed environment. In a spatially-structured environment, effects on self and on partner are both important. For example, a spatially-structured environment may favor an altruistic mutation that confers a large benefit on partner at a small cost to self. Parentheses indicate that selection outcome (favored or disfavored) depends on quantitative details of the fitness effects on self and partner (see *Momeni et al., 2013b* for an example). Note that a mutation that is strictly partner-serving or altruistic could still rise in frequency in a well-mixed environment by "hitchhiking" with other self-serving mutations (*Morgan et al., 2012*; *Waite and Shou, 2012*). (**C**) CoSMO. CoSMO is an engineered mutualistic community consisting of two non-mating *S. cerevisiae* strains (*Hart et al., 2019a*; *Shou et al., 2007*). Thus, the two strains may be regarded as two species. The mCherry-expressing $L^-H^+$ strain is unable to synthesize lysine (*L*) and overproduces the adenine precursor hypoxanthine (*H*). The complementary GFP-expressing $H^-L^+$ strain requires hypoxanthine and overproduces lysine. Both overproduction mutations render the first enzyme of the corresponding biosynthesis pathway insensitive to end-product feedback inhibition control (*Armitt and Woods, 1970*; *Feller et al., 1999*). In minimal medium lacking exogenously supplied *L* and *H*, the two strains form a mutualistic community where live cells from both strains release overproduced metabolites (*Hart et al., 2019a*) and support each other's growth.

DOI: https://doi.org/10.7554/eLife.44812.003

The following source data and figure supplements are available for figure 1:

**Figure supplement 1.** Lysine overproduction incurs a fitness cost.

DOI: https://doi.org/10.7554/eLife.44812.004

*Figure 1 continued*

**Figure supplement 1—source data 1.** Data plotted in *Figure 1—figure supplement 1*.
DOI: https://doi.org/10.7554/eLife.44812.005

mutualistic cooperation, especially given the genetic tractability of budding yeast and validated phenotype quantification methods that we have developed (*Hart et al., 2019a*; *Hart et al., 2019b*).

In this study, we demonstrate that an intuitive definition of partner-serving phenotype in mutualism can lead to erroneous conclusions. We will conclude by discussing how to quantify important theoretical concepts such as 'benefit', 'cost', and 'partner-serving phenotype', especially for microbial mutualisms where interactions span multiple generations.

## Results

We propagated nine independent CoSMO communities in well-mixed minimal medium without lysine or hypoxanthine supplements for ~100 generations by performing periodic dilutions to keep culture turbidity below saturation (Materials and methods, 'CoSMO evolution'). Due to the metabolic codependence, the two strains coexisted throughout evolution (*Shou et al., 2007*). We froze samples at various time points for later revival. Since all communities were well-mixed, we predicted that selfish, strictly self-serving, and win-win mutations should arise (*Figure 1B*). In this study, we focused on $L^-H^+$.

### Evolved $L^-H^+$ clones harboring Chromosome 14 duplication appeared to display a win-win phenotype

We randomly isolated evolved $L^-H^+$ clones from independent communities. Since the community environment was lysine-limited (*Hart et al., 2019a*; *Waite and Shou, 2012*), improved growth under lysine limitation would be self-serving. Indeed, while the ancestral strain failed to grow into microcolonies on agar with low lysine (1.5 µM), all tested (>20) evolved clones could (*Figure 2—figure supplement 1*; Materials and methods 'Microcolony assay'), consistent with our previous findings (*Hart et al., 2019a*; *Waite and Shou, 2012*). Thus, evolved $L^-H^+$ clones displayed self-serving phenotypes.

Since hypoxanthine is also scarce in the community (*Hart et al., 2019a*), a mutant $L^-H^+$ cell with increased hypoxanthine release rate should allow the partner to grow faster and is thus partner-serving. We randomly chose several evolved $L^-H^+$ clones, and measured their hypoxanthine release rate (*Hart et al., 2019a*) (Materials and methods, 'Release assay'; *Figure 3—figure supplement 1*). Whereas two of the evolved clones released hypoxanthine at a similar rate as the ancestor, three increased release rates in the sense that each cell released more hypoxanthine per hour than the ancestor (*Figure 3—figure supplement 2*). We then sequenced genomes of both types of clones (Materials and methods, 'Whole-genome sequencing'; *Figure 2—figure supplement 2*). Chromosome 14 duplication (*DISOMY14*) occurred in all three clones that exhibited faster-than-ancestor hypoxanthine release rate (*Figure 3—figure supplement 2*, blue), and not in the other two clones that exhibited ancestral release rate (*Figure 3—figure supplement 2*, orange).

When we back-crossed evolved clones harboring *DISOMY14* to the ancestral background (Materials and methods, 'Strains and medium'), only meiotic segregants containing *DISOMY14* showed increased hypoxanthine release rate compared to the ancestor (*Figure 3—figure supplement 3*). Thus, *DISOMY14* genetically co-segregated with increased release rate (partner-serving). *DISOMY14* repeatedly rose to a detectable frequency when $L^-H^+$ evolved with $H^-L^+$ in CoSMO (3 out of 3 lines), or when $L^-H^+$ evolved alone in lysine-limited chemostats (5 out of 5 lines) (*Supplementary file 3*). Thus, *DISOMY14* is likely adaptive in lysine limitation. Indeed, while ancestors failed to form microcolonies on low-lysine plate, meiotic segregants containing *DISOMY14* could (self-serving) (*Figure 2—figure supplement 1*). Taken together, we hypothesized *DISOMY14* to be both self-serving and partner-serving, that is 'win-win'.

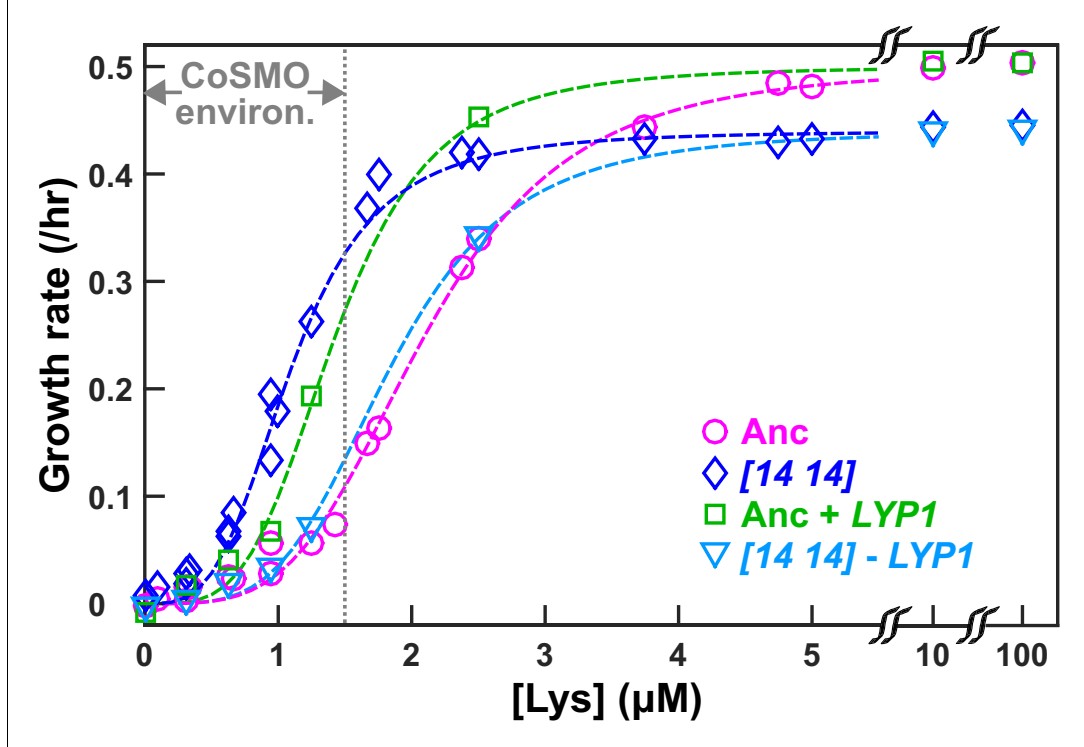

**Figure 2.** *DISOMY14* improves $L^-H^+$ growth rate under lysine limitation via duplication of the high-affinity lysine permease *LYP1*. Exponentially-growing cells were washed free of lysine, starved in minimal medium SD for 3~5 hrs to deplete vacuolar lysine storage, and incubated in microtiter wells containing SD supplemented with various concentrations of lysine. Cells were imaged using a fluorescence microscope, and total fluorescence was tracked over time (Materials and methods; "Microscopy growth assay") (*Hart et al., 2019b*). The maximal growth rate (the steepest positive slope of ln [fluorescence] against time) was quantified and plotted against lysine concentration. The grey dotted line demarcating "CoSMO environ." corresponds to the lysine level supporting a growth rate of $\leq 0.1$/hr as observed in ancestral CoSMO. *DISOMY14* ("*[14 14]*", blue diamond; WY2261) grew faster than the ancestral $L^-H^+$ (magenta circle; WY1335) in low lysine. This self-serving phenotype of *DISOMY14* was abolished when the duplicated *LYP1* gene was deleted form *DISOMY14* (cyan triangle; WY2262, WY2263). Conversely, introducing an extra copy of *LYP1* into the ancestor improved grow rate in limited lysine (green square; WY2254, WY2255). Each data point is the average of multiple (~4) experiments. Dashed fitting curves: Moser growth equation $g = g_{max}L^n/(K^n + L^n)$ where $g$ is growth rate of $L^-H^+$, $L$ is lysine concentration, $g_{max}$ is the maximal growth rate in excess lysine, $K$ is the lysine concentration supporting $g_{max}/2$, and $n$ is the "growth cooperativity" constant describing the sigmoidal shape of the curve. The maximal growth rate of *DISOMY14* is lower than that of the ancestor, which could be due to the fitness cost associated with aneuploidy (*Oromendia et al., 2012*). Data can be found in *Figure 2—source data 1*.

DOI: https://doi.org/10.7554/eLife.44812.006

The following source data and figure supplements are available for figure 2:

**Source data 1.** Data plotted in *Figure 2*.
DOI: https://doi.org/10.7554/eLife.44812.009
**Figure supplement 1.** Micro-colony assay distinguishes ancestral from evolved $L^-H^+$ clones.
DOI: https://doi.org/10.7554/eLife.44812.007
**Figure supplement 2.** Whole genome sequencing Nextera V2.
DOI: https://doi.org/10.7554/eLife.44812.008

## The self-serving phenotype of *DISOMY14* requires duplication of the lysine permease *LYP1*

Chromosome 14 harbors the high-affinity lysine permease *LYP1*. To test whether *LYP1* duplication might improve the growth rate of $L^-H^+$ in limited lysine, we inserted an extra copy of *LYP1* into the ancestral $L^-H^+$ strain (Materials and methods, 'Gene knock-in and knock-out'), and quantified cell growth rate under various concentrations of lysine using a microscopy assay (Materials and methods, 'Microscopy growth assay') (*Hart et al., 2019b*). *LYP1* duplication indeed significantly increased the growth rate of $L^-H^+$ in low lysine (*Figure 2*, compare green with magenta). Similarly, deleting the duplicated copy of *LYP1* from *DISOMY14* cells abolished the self-serving phenotype of *DISOMY14*

(*Figure 2*, compare blue and cyan with magenta). Taken together, duplication of *LYP1* is responsible for the self-serving phenotype of *DISOMY14*.

## *WHI3* duplication is responsible for the increased hypoxanthine release rate of *DISOMY14*

To identify which duplicated gene(s) might be responsible for the increased hypoxanthine release rate of *DISOMY14*, we systematically deleted various sections of the duplicated Chromosome 14 (*Figure 3A*; Materials and methods, 'Chromosome truncation'), and quantified hypoxanthine release rate (*Figure 3B*). Duplication of the region between *YNL193W* and *GCR2* was necessary for the increased release rate (*Figure 3B*, orange). This region contains six genes, including *WHI3*. Integrating an extra copy of *WHI3* into the ancestor increased release rate to near that of *DISOMY14*, while deleting one copy of *WHI3* from *DISOMY14* restored the ancestral release rate (*Figure 4A*; Materials and methods, 'Gene knock-in and knock-out').

## Despite increased release rate per cell, *DISOMY14* is not partner-serving during steady state community growth

*WHI3* encodes an inhibitor of the cell division cycle (*Garí et al., 2001*; *Nash et al., 2001*). When *WHI3* is overexpressed, cell division becomes less frequent even though biomass grows at the same rate, resulting in larger cells. Consistent with this notion, deletion of *WHI3* results in smaller cell size, whereas extra copies or overexpression of *WHI3* increases cell size (*Garí et al., 2001*; *Nash et al., 2001*). Indeed, *DISOMY14* cells are bigger than ancestral cells, as quantified by the Coulter counter (*Figure 4—figure supplement 1A*; Materials and methods, 'Cell size measurements'). Integrating an extra copy of *WHI3* into the ancestor increased mean cell size, while deleting the extra copy of *WHI3* from *DISOMY14* restored ancestral cell size (*Figure 4—figure supplement 1A*).

Consistent with *DISOMY14* cells being bigger than ancestral cells, *DISOMY14* cells utilized more lysine per birth than the ancestor (*Figure 4B*; Materials and methods, 'Metabolite utilization in batch culture'). Integrating an extra copy of *WHI3* into the ancestor increased lysine utilization per newborn cell, while deleting the extra copy of *WHI3* from *DISOMY14* reduced lysine utilization per newborn cell (*Figure 4—figure supplement 1B*). Thus, although each *DISOMY14* cell released more hypoxanthine, each also utilized more lysine.

When we normalized hypoxanthine release rate per $L^-H^+$ cell ($r_H$) by lysine utilization per $L^-H^+$ birth ($u_L$), which we define as the '$H$-$L$ exchange ratio' ($r_H/u_L$), *DISOMY14* is indistinguishable from the ancestor (*Figure 4C*). In other words, a fixed amount of lysine can be converted to ancestral $L^-H^+$ cells which are more numerous but lower-releasing (*Figure 5B,i*), or *DISOMY14* cells which are fewer but higher-releasing (*Figure 5B,ii*). As long as the ancestor and *DISOMY14* displayed a similar *total* hypoxanthine release rate per lysine utilized, the partner $H^-L^+$ would not benefit more from one than the other during long-term community growth. In Discussion, we will describe how exchange ratio links to concepts such as 'inclusive fitness'.

If *DISOMY14* is not partner-serving, then fixed lysine supply would lead *DISOMY14* and ancestor to release identical total hypoxanthine. To test this hypothesis, we cultured ancestral and *DISOMY14* $L^-H^+$ cells in lysine-limited chemostats with a fixed rate of lysine supply (Materials and methods, 'Chemostat'). Indeed, the steady state hypoxanthine concentrations in chemostats were indistinguishable between ancestor and *DISOMY14* (*Figure 4D*). Similar to batch culture results, in chemostats *DISOMY14* released hypoxanthine at a higher rate than the ancestor on a per cell basis, but utilized more lysine per birth, leading to an identical $H$-$L$ exchange ratio as the ancestor (*Figure 4—figure supplement 2B*).

Consistent with *DISOMY14* not being partner-serving, partner $H^-L^+$ grew at the same steady state rate whether co-cultured with ancestral or *DISOMY14* $L^-H^+$ (*Figure 4E*, right panel). This also suggests that no new beneficial interactions have evolved between *DISOMY14* $L^-H^+$ and partner $H^-L^+$. Note that CoSMO grew at a steady state rate after an initial lag (*Figure 4—figure supplement 3*) (*Hart et al., 2019a*). Here we do not consider the duration or growth rate during the lag phase, but rather, long-term steady state growth rate.

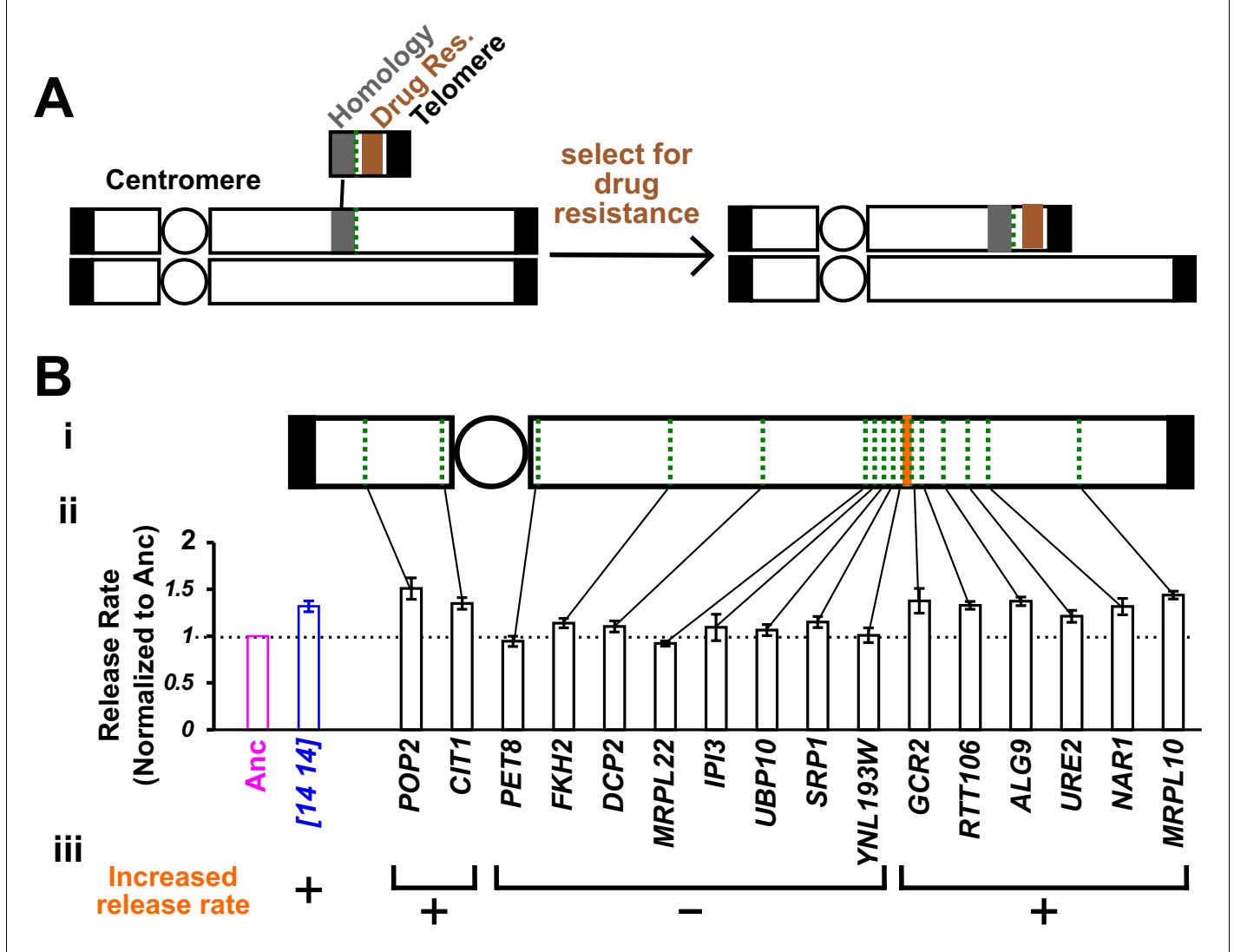

**Figure 3.** Chromosome truncation reveals the genomic region responsible for increased hypoxanthine release rate. (**A**) Chromosome truncation scheme. Recombination occurs between a chromosome and a truncation cassette containing a homology region, a drug resistance marker, and a telomere (Materials and methods, 'Chromosome truncation'). Single integration leads to the truncation of one of the duplicated chromosomes, with the centromere (circle)-containing region being retained by the cell. Integration into both chromosomes would lead to an inviable cell due to loss of the segment immediately distal to the insertion site from both chromosomes (not drawn). The copy number of chromosomal regions in transformants was verified by RADseq (Materials and methods, 'RADseq'; *Figure 3—figure supplement 4*). (**B**) The genomic region between *YNL193W* and *GCR2* is responsible for increased hypoxanthine release rate. We systematically truncated chromosome 14 in *DISOMY14* cells (WY2261). We chose truncation sites (green dotted lines) that were spread across chromosome 14, and in later rounds of truncation, spread across the region of interest (**i**). We quantified hypoxanthine release rates of transformants in starvation batch cultures (Materials and methods, 'Release assay'; *Figure 3—figure supplement 1*), normalized them against the release rate of the ancestral strain measured in the same experiment, and plotted the mean value and two standard errors of mean (**ii**). Duplication of the region between *YNL193W* and *GCR2* (orange; containing six genes including *YNL193W* and excluding *GCR2*) is responsible for the increased hypoxanthine release rate (**iii**). Data can be found in *Figure 3—source data 1*.
DOI: https://doi.org/10.7554/eLife.44812.010

The following source data and figure supplements are available for figure 3:

**Source data 1.** Data plotted in *Figure 3*.
DOI: https://doi.org/10.7554/eLife.44812.018

**Figure supplement 1.** Starvation batch culture release assay.
DOI: https://doi.org/10.7554/eLife.44812.011

**Figure supplement 2.** Evolved $L^-H^+$ clones with improved hypoxanthine release rates harbored Chromosome 14 duplication.
DOI: https://doi.org/10.7554/eLife.44812.012

*Figure 3 continued on next page*

*Figure 3 continued*

**Figure supplement 2—source data 1.** Data plotted in *Figure 3—figure supplement 2*.

DOI: https://doi.org/10.7554/eLife.44812.013

**Figure supplement 3.** Increased hypoxanthine release rate segregates with *DISOMY14*.

DOI: https://doi.org/10.7554/eLife.44812.014

**Figure supplement 3—source data 2.** Data plotted in *Figure 3—figure supplement 3*.

DOI: https://doi.org/10.7554/eLife.44812.015

**Figure supplement 4.** RADseq.

DOI: https://doi.org/10.7554/eLife.44812.016

**Figure supplement 5.** Bioassay.

DOI: https://doi.org/10.7554/eLife.44812.017

Taken together, despite having a higher hypoxanthine release rate per cell, *DISOMY14* is not partner-serving or win-win during steady state community growth. Rather, *DISOMY14* is strictly self-serving.

## Discussion

Interpreting social phenotypes can be surprisingly nuanced (*Dubravcic et al., 2014*; *Gore et al., 2009*; *Greig and Travisano, 2004*; *Kümmerli and Ross-Gillespie, 2014*; *Rainey et al., 2014*; *smith et al., 2014*; *Zhang and Rainey, 2013*). Here, by dissecting the molecular bases of how *DIS-OMY14* affects self ($L^-H^+$) and partner ($H^-L^+$), we illustrate how microbial win-win or partner-serving phenotypes might be interpreted.

The self-serving aspect of *DISOMY14* is straightforward: due to duplication of the lysine permease gene *LYP1*, *DISOMY14* grows faster than the ancestor in lysine limitation typically found in the CoSMO environment (*Figure 2*).

The partner-serving aspect of *DISOMY14* is far less straightforward. We had mis-interpreted *DIS-OMY14*'s increased hypoxanthine release rate per cell as partner-serving until we realized that duplication of *WHI3*, a cell division cycle inhibitor that makes cells larger, was responsible. Thus, when quantifying partner-serving phenotypes in microbial mutualisms that span multiple generations, we may need to consider exchange ratio (benefit release rate per cell divided by benefit utilized to make that cell, or equivalently as we will show later, total benefit release rate per benefit utilized). Below, we discuss how exchange ratio links to current frameworks of mutualisms, inclusive fitness theory, and biological market theory.

### Theories of mutualisms

General theories have been developed for mutualisms (*Archetti et al., 2011*; *Doebeli and Knowlton, 1998*; *Foster and Wenseleers, 2006*; *Frank, 1994*; *Jones et al., 2015*; *Sachs et al., 2004*; *Trivers, 1971*; *West et al., 2002*; *Yamamura et al., 2004*). In mathematical models of mutualisms (e.g. *Doebeli and Knowlton, 1998*; *Foster and Wenseleers, 2006*; *Frank, 1994*; *Yamamura et al., 2004*), exchanged goods (investments) were linked to fitness effects on the focal individual. For example, the net fitness gain of a focal individual = fitness gain per investment made * average investment made within the group – fitness loss per investment made * investment by the focal individual (*Frank, 1994*). For a focal $H^-L^+$, the fitness loss term (fitness loss per lysine released*total lysine released by the focal cell) is fixed whether the interaction partner is ancestral or *DISOMY14* $L^-H^+$. The term of fitness gain per investment made (how much faster $H^-L^+$ would grow per unit of lysine supplied to $L^-H^+$) does embody the spirit of exchange ratio (total hypoxanthine return rate per unit of lysine invested). However, measuring fitness gain per investment can be difficult since this value is unlikely to be a constant. For example, the growth rate of $L^-H^+$ is not a linear function of lysine concentration (*Figure 2*). In contrast in exchange ratio, 'benefit' and 'investment' are defined in physical terms of the goods exchanged, and fitness 'cost' for making the investment is separately considered (see *Equation 1* below).

A focal $L^-H^+$'s exchange ratio is equivalent to partner's benefit-to-investment ratio. There are in fact two perspectives for this equivalence. In the 'individualistic' perspective, we can define the benefit gained by partner $H^-L^+$ as the hypoxanthine release rate by the focal $L^-H^+$ cell ($r_H$),

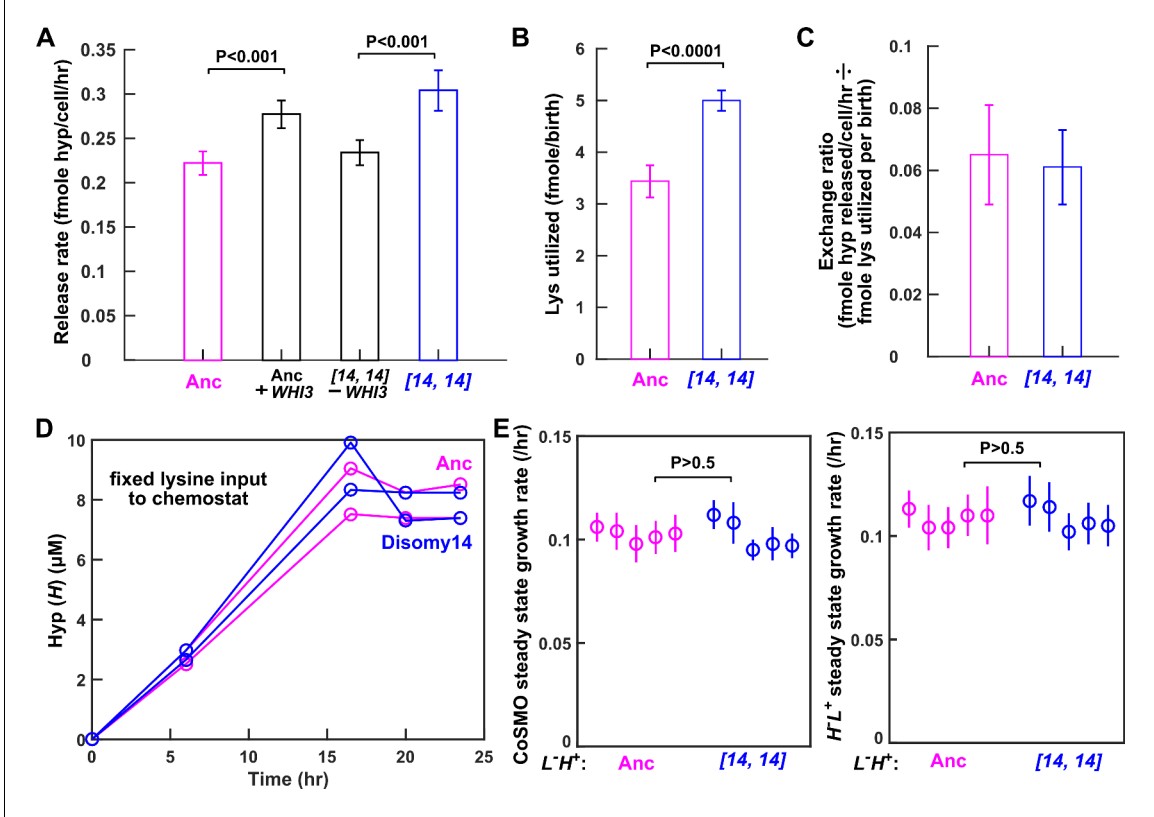

**Figure 4.** Despite increased release rate per cell, *DISOMY14* is not partner-serving. (A) *WHI3* duplication is responsible for increased hypoxanthine release rate per cell. Introducing an extra copy of *WHI3* into the ancestral background (WY2357 ~2359) increased hypoxanthine release rate per cell. Deleting the duplicated *WHI3* from *DISOMY14* (WY2350 ~2352) decreased release rate (Materials and methods, 'Release rate'; starvation batch culture). (B) *DISOMY14* utilizes more lysine per birth than the ancestor (Materials and methods, 'Metabolite utilization in batch culture'). For A and B, mean and two SEM are plotted. (C) Indistinguishable exchange ratio between ancestor and *DISOMY14*. Hypoxanthine release rate per $L^-H^+$ cell (A) is divided by lysine utilization per $L^-H^+$ birth (B). Error bars are calculated via error propagation (Materials and methods, 'Calculating uncertainty of ratio'). (A–C) were measured in starvation batch cultures. (D) In chemostats fed with a fixed lysine supply rate, *DISOMY14* and ancestor release the same total hypoxanthine. Ancestor (WY1335, magenta) or *DISOMY14* (WY2349, blue) $L^-H^+$ cells were grown in lysine-limited chemostats with a doubling time of 6 hr (Materials and methods, 'Chemostat'). Supernatant hypoxanthine concentrations were quantified using a yield-based bioassay (Materials and methods, 'Bioassay'; *Figure 3—figure supplement 5*). For complete data and statistical comparison, see *Figure 4—figure supplement 2*. (E) *DISOMY14* and ancestral $L^-H^+$ led to identical growth rate of community and of partner. To prevent rapid evolution in $L^-H^+$ (*Hart et al., 2019a*), we grew CoSMO in a spatially-structured environment on agarose pads (Materials and methods, 'Community growth rate'), and periodically measured the absolute abundance of the two strains (differentiable by their fluorescence; Materials and methods, 'Flow cytometry'). We then quantified the steady state growth rate of community and of partner $H^+L^-$ by regressing ln(cell density) against time after the initial lag phase up to <10[8] cells (*Hart et al., 2019a*; see *Figure 4—figure supplement 3* for examples of detailed dynamics). We plotted the slope (i.e. growth rate), with error bars indicating 2x standard error of estimating the slope. In A, B, and E, we performed statistical comparisons first using the F-test to test for equal variance, and then using unpaired two-tailed t-test with equal variance. We plotted the corresponding P-values of the t-test (the probability of observing a test statistic as extreme as, or more extreme than, the observed value under the null hypothesis of two groups belonging to the same distribution). Comparisons in A and B show significant difference, while those in E are not significantly different. All data can be found in *Figure 4—source data 1*.
DOI: https://doi.org/10.7554/eLife.44812.019

The following source data and figure supplements are available for figure 4:

**Source data 1.** Data plotted in *Figure 4*.
DOI: https://doi.org/10.7554/eLife.44812.023

**Figure supplement 1.** *WHI3* duplication increases cell size and lysine utilization per birth.
DOI: https://doi.org/10.7554/eLife.44812.020

**Figure supplement 2.** In chemostats, *DISOMY14* and ancestor share identical exchange ratio.
DOI: https://doi.org/10.7554/eLife.44812.021

**Figure supplement 3.** CoSMO growth dynamics.
DOI: https://doi.org/10.7554/eLife.44812.022

while investment made by $H^-L^+$ as the amount of lysine required to make the focal $L^-H^+$ cell ($u_L$). Then, $H^-L^+$'s benefit-to-investment ratio is identical to $L^-H^+$'s exchange ratio. In the 'population' perspective, we can define the investment made by $H^-L^+$ as a unit of lysine released for $L^-H^+$, and the benefit received by $H^-L^+$ as the resultant *total* rate of hypoxanthine reciprocation from not only the focal $L^-H^+$ cells but also its offspring (*Figure 5B*). This benefit-to-investment ratio is also identical to the exchange ratio, since total hypoxanthine release rate/fmole lysine = (hypoxanthine release rate per cell * total number of $L^-H^+$ cells)/fmole lysine =$r_H$/(fmole lysine/total number of $L^-H^+$ cells)=$r_H/u_L$.

## Inclusive fitness

Why would an individual cooperate – paying a fitness cost to provide a benefit that can aid the reproduction of other individuals (e.g. sterile ant workers aiding the reproduction of the queen)? Social evolution theories offer explanations for the evolution of cooperative traits (*Frank, 1998*; *Hamilton, 1964*; *Kerr, 2009*; *Lehmann and Keller, 2006*; *Maynard Smith, 1964*; *Price, 1972*; *Price, 1970*; *Queller, 1985*; *Sachs et al., 2004*; *Traulsen and Nowak, 2006*; *West et al., 2007*).

A central concept in social evolution theories is 'inclusive fitness'. Inclusive fitness considers the fitness impact from social interactions, and when properly formulated, natural selection leads organisms to become adapted as if to maximize their inclusive fitness (*Grafen, 2006*). For example, Hamilton's rule states that cooperation can evolve as long as $rb$-$c$ >0, where $c$ is the fitness cost to the focal cooperator, $b$ is the benefit to focal cooperator's partner, and $r$ is their 'relatedness' – the similarity of an actor to its recipient relative to the population (*Damore and Gore, 2012*; *Fletcher and Doebeli, 2009*; *Fletcher and Doebeli, 2009*; *Hamilton, 1964*; *Queller, 1992*; *van Veelen, 2009*). Note that 'similarity' can broadly refer to action type (e.g. cooperation versus no cooperation), even if the actor and the recipient are genetically unrelated as in mutualisms. A mathematically-equivalent, individual-centric version of inclusive fitness (also known as 'direct fitness') of a focal individual is the sum of its basal fitness $w_0$ in an asocial environment plus $rb$ benefit received from other cooperators in the social environment minus the cost of cooperation $c$ (*Damore and Gore, 2012*; *Foster and Wenseleers, 2006*; *Frank, 1998*; *West et al., 2007*). If a focal individual's inclusive fitness is greater than its basal fitness ($w_0$+$rb$-c>$w_0$ or $rb$-$c$ >0), then the individual would grow faster by cooperating than by not cooperating. Inclusive fitness relies on assumptions such as additivity of fitness effects. However, such assumptions are often not valid in microbial communities. For example, fitness often increases in a nonlinear (e.g. sigmoidal or saturating) fashion as the benefit increases (*Figure 2*; *Niehaus et al., 2019*), and therefore inclusive fitness described above is

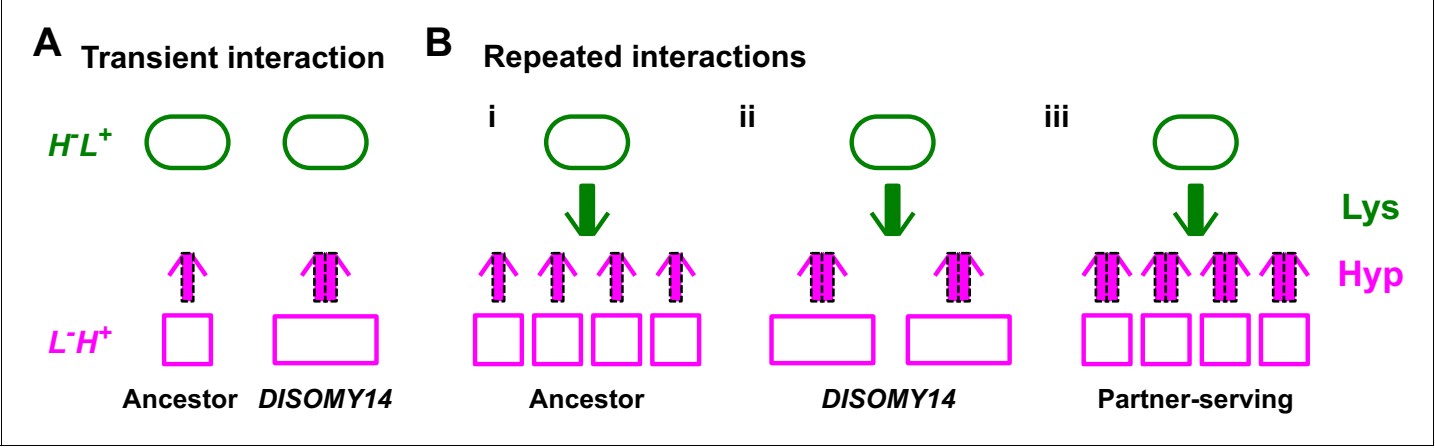

**Figure 5.** Schematic illustration of partner-serving mutations. (**A**) Transient interaction. At the initial stage of an interaction, *DISOMY14* with an increased release rate per cell (2x thick purple arrow) is more partner-serving than the ancestor (1x thick purple arrow). (**B**) Long-term repeated interactions. Over multiple generations, *DISOMY14* is not partner-serving. Even though each *DISOMY14* cell releases more hypoxanthine than each ancestral cell, a fixed amount of lysine (green arrow) is turned into more ancestral cells than *DISOMY14* cells (compare i and ii). A true partner-serving mutation should increase exchange ratio (iii), which is expressed as total benefit supply rate per intake benefit, or (benefit supply rate per cell)/(benefit amount utilized to make the cell).
DOI: https://doi.org/10.7554/eLife.44812.024

thought to be over-simplified and sometimes even misleading (*Damore and Gore, 2012*; *Grafen, 2006*; *Nowak et al., 2010*; *van Veelen, 2009*). However, despite these criticisms, inclusive fitness has served as a useful conceptual framework for many (*Abbot et al., 2011*; *Birch, 2017*).

In obligatory mutualisms between clonal populations (i.e. relatedness = 1), we arrive at a physical definition of inclusive fitness (or direct fitness) based on exchange ratios and costs of making investments. Specifically, the fitness of $H^-L^+$ and $L^-H^+$ in monoculture is negative due to death rate and the cost of making investments. When growing in communities, divergent strain ratios rapidly converge to a fixed value (*Momeni et al., 2013a*; *Shou et al., 2007*). This means that the two strains must grow at an identical rate which equals to the community growth rate $g_{comm}$. In other words, the growth rate of the two strains are identical to each other and to community growth rate (*Hart et al., 2019a*; Materials and methods, 'Community growth rate'):

$$g_{comm} = -\frac{(d_H + c_H + d_L + c_L)}{2} + \sqrt{\frac{r_H r_L}{u_L u_H} + \frac{(d_H + c_H - d_L - c_L)^2}{4}} \qquad (1)$$

Here, $d_H$ and $d_L$ are respectively $H^-L^+$ and $L^-H^+$'s death rates, $c_H$ and $c_L$ are respectively $H^-L^+$ and $L^-H^+$'s fitness costs of overproducing metabolites, $r_H$ and $r_L$ are respectively hypoxanthine and lysine release rates, and $u_H$ and $u_L$ are respectively hypoxanthine and lysine utilization amount per birth. Note that $\sqrt{\frac{r_H r_L}{u_L u_H}}$ is the geometric mean of the two exchange ratios $r_H / u_L$ and $r_L / u_H$, and $c_H$ and $c_L$ represent the fitness costs of making investments for partner.

A corollary to *Equation 1* is that despite *DISOMY14*'s better affinity for metabolite (*Figure 2*), *DISOMY14* and ancestral $L^-H^+$ have identical fitness when cocultured with $H^-L^+$ in separate CoSMO communities (*Figure 4E*). This is because in CoSMO, the growth of an individual is limited by partner-supplied metabolites. Thus, there is no point of eating faster if meals arrive slowly. Obviously, if we were to culture *DISOMY14* $L^-H^+$, ancestral $L^-H^+$, and $H^-L^+$ in a single community in a well-mixed environment, then *DISOMY14* should outcompete ancestor due to its better affinity for lysine (*Figure 2*).

This brings up an important point: fitness metric based on exchange ratio only works for cases where each mutualistic partner is of one genotype. It does not capture changes in genotype frequency within a species. Because of this restriction, it is best suited for sub-communities in a spatially-structured environment. In a spatially-structured environment, if the initial population densities are low, then interactions will mainly take place between one genotype from each species. Individuals in the local sub-community with the highest $g_{comm}$ are predicted to grow the fastest. In the case of spatial CoSMO, would an $H^-L^+$ cell grow faster if it had landed next to an ancestral or a *DISOMY14* $L^-H^+$ cell? During the initial encounter, $H^-L^+$ next to *DISOMY14* will benefit more from *DISOMY14*'s faster release rate (*Figure 5A*). However, after this initial stage, *DISOMY14* and ancestor are identical due to identical exchange ratio (*Figure 5B,i and ii*). Indeed, $H^-L^+$ grew equally fast when co-cultured with ancestral or *DISOMY14* $L^-H^+$ (*Figure 4E*).

## Biological market theory

Biological market theory posits that the exchange of goods among organisms can be analyzed in market terms, where individuals attempt to maximize their gains (*Noë and Hammerstein, 1994*; *Noë and Hammerstein, 1995*; *Werner et al., 2014*). For example, a male insect that offers more nuptial gifts to a female is regarded by the female as being more partner (female)-serving and is thus chosen by the female (*Noë and Hammerstein, 1994*; *Noë and Hammerstein, 1995*; *Werner et al., 2014*). Although biological market theory does not apply here since our yeast strains lack partner choice capability, our work can prove useful for other systems. For example, consider the legume-rhizobia mutualism where a legume host provides photosynthates to rhizobia while rhizobia reciprocate fixed nitrogen. Application of biological market theory leads to statements such as "...whether plant hosts can detect variation in resources or services provided (by rhizobia) and respond accordingly. Such discrimination mechanisms have been found in legumes, with some species preferentially supporting rhizobial symbionts that provide more fixed $N_2$ for hosts" (*Werner et al., 2014*). Since legume-rhizobia mutualisms last over multiple generations of rhizobia, we suggest that 'fixed $N_2$' should be quantified in terms of exchange ratio: a focal rhizobium's nitrogen release rate normalized by photosynthate utilized to make the rhizobium, or total

nitrogen release rate per photosynthate utilized. Obviously, when comparing non-nitrogen fixers with nitrogen fixers, fixers are more mutualistic than non-fixers. But when comparing quantitative variants of nitrogen fixers, this distinction could become important. By explicitly quantifying the exchanged goods, exchange ratio captures the spirit of market economy.

In summary, when considering microbial mutualisms that span multiple generations, a focal individual's partner-serving phenotype can be quantified as the exchange ratio of benefit production rate by the focal individual for partner divided by benefit utilized to make the focal individual. This is equivalent to total benefit production rate for partner per intake benefit. By the same token, an individual with reduced benefit production rate may not be a cheater if the individual also utilizes less benefit from the partner.

For a lay audience behind-the-scenes story, see 'Foresight, hindsight, insight, and the blur in-between' (*Supplementary file 1*; https://medium.com/@wenying.shou/foresight-hindsight-insight-and-the-blur-in-between-9dd0505b2918).

## Materials and methods

### Strains and medium

Genetic manipulations and growth medium for the yeast *S. cerevisiae* are explained in *Guthrie and Fink (1991)*. Protocols and technical details that we have used can be found in *Waite and Shou (2014)*. Briefly, we used autoclaved rich medium YPD (10 g/L yeast extract, 20 g/L peptone, 20 g/L glucose) in 2% agar plates for isolating single colonies. Saturated YPD overnight liquid cultures from these colonies were then used as inocula to grow exponential cultures. YPD overnight cultures were stored at room temperature for no more than 4 ~ 5 days prior to experiments.

We used defined minimal medium SD (6.7 g/L Difco yeast nitrogen base with ammonium sulfate without amino acids, 20 g/L glucose) for all experiments, with supplemental metabolites as noted (*Guthrie and Fink, 1991*). To achieve higher reproducibility, we sterilized SD media by filtering through 0.22 µm filters. To make SD plates, we autoclaved 20 g/L Bacto agar or agarose in $H_2O$, and after autoclaving, supplemented equal volume of sterile-filtered 2XSD.

The ancestral strain was WY1335, described in detail in *Hart et al. (2019a)*. All strains are in *Supplementary file 2*.

The *DISOMY14* strain we used for analysis WY2261 (refrozen as WY2348 and WY2349) was obtained in the following manner. The evolved strain WY1584 was back-crossed twice into the ancestral background to get rid of mutations in genes *ECM21* and *YPL247C*. The first cross with WY1521 resulted in '38-1D', which was then crossed with WY1335 to result in WY2261 ('E2'). To genotype spores, we PCR amplified the mutated regions in *ECM21* and *YPL247C*, and subjected the purified PCR product to Sanger sequencing. For those spores that contained no mutations in *ECM21* and *YPL247C*, we subjected them to restriction-site associated DNA sequencing (Materials and methods, 'RADseq') to determine ploidy. When we modified our sequence analysis pipeline, we realized that WY2261 contained other mutations (*Supplementary file 3*). However, the presence of other mutations does not affect our conclusions, since integrating an extra copy of *LYP1* or *WHI3* into the ancestral background respectively increased growth rate under lysine limitation (*Figure 2*) and per cell hypoxanthine release rate (*Figure 4A*).

### CoSMO evolution

$L^-H^+$ (WY1335) and $H^-L^+$ (WY1340) were grown separately to exponential phase in minimal SD medium supplemented with lysine (164.3 µM) or adenine sulfate (108.6 µM), respectively (*Guthrie and Fink, 1991*). Cells were washed free of supplements, counted using a Coulter counter, and mixed at 1000:1 (Line A), 1:1 (Line B), or 1:1000 (Line C) at a total density of $5 \times 10^5$/ml. The different initio ratios did not noticeably affect evolutionary outcomes. Three 3 ml community replicates (replicates 1, 2, and 3) per initial ratio were initiated. Communities were grown at 30°C in glass tubes on a rotator to ensure well-mixing. Community turbidity was tracked by measuring the optical density ($OD_{600}$) in a spectrophotometer once to twice every day. In this study, 1 OD was found to be $2 \sim 4 \times 10^7$ cells/ml. We diluted communities periodically to maintain OD at below 0.5 to avoid additional selections due to limitations of nutrients other than hypoxanthine or lysine. The fold-dilution

was controlled to within 10 ~ 20 folds to minimize severe population bottlenecks. Note that no mutagens were used during evolution.

Coculture generation was calculated from accumulative population density by multiplying OD with total fold-dilutions. For each coculture at every 10 ~ 20 generations, cell pellet of ~1 ml coculture was resuspended in 1 ml rich medium YPD (*Guthrie and Fink, 1991*)+10% trehalose, cooled at 4°C for several hours, and frozen at −80°C. Cells frozen this way revived much better than if frozen in SD medium supplemented with a final of 15% glycerol.

CoSMO could engage in self-sustained growth only if its initial total cell density was sufficiently high (*Shou et al., 2007*). Thus, to revive a coculture, ~20 μl was scooped from the frozen stock using a sterile metal spatula, diluted ~10 fold into SD, and allowed to grow to moderate turbidity. The coculture was further expanded by adding 3 ml of SD. To isolate clones, cocultures were plated on rich medium YPD, and clones from the two strains were distinguished by their fluorescence colors or drug resistance markers.

## Gibson assembly

The detailed protocol of Gibson assembly (*Gibson et al., 2009*) for assembling DNA fragments with end homology was obtained from Eric Klavins lab (University of Washington). 1 ml 5xISO buffer: 1M Tris-HCl (pH 7.5) 500 μl; 2M MgCl₂ 25 μl; 100 mM dGTP, dATP, dCTP, and dTTP 10 μl each (total 40 μl); 1M DTT 50 μl; 100 mM NAD 50 μl; PEG-8000 0.25 g; H₂O: 145 μl. 5xISO was frozen in 100 μl aliquots at −20°C.

The assembly master mix (375 μl total) included: H₂O 216.75 μl; 5XISO Buffer 100 μl; 1 U/μl T5 Exonuclease 2 μl; 2 U/μl Phusion Polymerase 6.25 μl; 40 U/μl Taq DNA Ligase 50 μl. 15 μl aliquots were stored at −20°C. This master mix is ideal for DNA molecules with 20 ~ 150 bp overlapping homology. To carry out the assembly reaction, 15 μl assembly master mix is mixed with a total of 5 μl DNA (e.g. 125 nM), and incubated at 50°C for 1 hr.

## Chromosome truncation

We constructed plasmid WSB175 to contain G418 resistance (*KanMX*) and a telomeric sequence. Briefly, WSB174 (from Dan Gottschling lab) containing a *URA3* marker was digested with HindIII and BamHI to remove the *URA3* marker and yield the vector backbone (4.5 kb). The *KanMX* gene was amplified from WSB26 using primers WSO433 and WSO434, each containing a 25 bp overhang homologous to the vector backbone. The vector backbone and the *KanMX* PCR product were circularized via Gibson assembly to yield WSB175. To perform Gibson assembly, we used 5 μl DNA (including 55 ng vector and 0.5 μl of 125 nM insert). Gibson mixture was transformed into *E. coli*, and DNA was extracted from several colonies. DNA was checked via restriction digestion with HindIII and BamHI.

To carry out chromosome truncation, we PCR amplified ~600 bp fragments from various genomic locations on Chromosome 14 (*Figure 3B*). The PCR reaction consisted of: genomic DNA (0.5 μl out of 25 μl where 0.3 ml overnight yeast culture was harvested and DNA extracted), 10xPCR buffer (5 μl), 25 mM MgCl₂ (3 μl), 10 mM dNTP mix (1 μl), 50 μM forward primer (0.5 μl), 50 μM reverse primer (0.5 μl), Taq polymerase (0.5 μl), H₂O (39 μl). Cycling conditions are as following: 94°C 3 min; [94°C 30 s +56.9°C 60 s + 72°C 60 s] x30 cycles; 72°C 10 min. We used Qubit (Thermofisher) to quantify DNA concentration. We then assembled PCR fragment with vector backbone containing *KanMX* and telomere (2.5 μl of each at 125 nM, corresponding to ~100 ng insert and 350 ng vector) to yield the assembled DNA. The assembled DNA (100 ng) was PCR amplified again. PCR reaction contained: Gibson product (4.1 μl), 5xPhusion HF buffer (10 μl), 10 mM dNTP mix (1 μl), 50 μM forward primer (1 μl), 50 μM reverse primer (WSO157, 1 μl), Phusion polymerase (0.5 μl), H₂O (32.4 μl). Cycling conditions are as following: 98°C 30 s; [98°C 10 s +59.9°C 30 s + 72°C 90 s] x30 cycles; 72°C 10 min. Expected length was ~2.4 kb.

The PCR product was used to transform the *DISOMY14* yeast strain (WY2261) using lithium acetate yeast transformation (*Waite and Shou, 2014*). Transformants were selected on rich medium supplemented with G418 (YPD + G418) plate. Transformants were screened for correct integration using PCR amplification across the chromosome integration site (one primer homologous to the genome, and the other primer homologous to *KanMX*).

## Gene knock-in and knock-out

To introduce an extra copy of *LYP1* in the *ste3::HygMX* locus of the ancestor, we assembled and transformed the following. We amplified a 527 bp homology region upstream of *STE3*, the *LYP1* gene (including 333 bp upstream and 466 bp downstream of ORF), and the *KanMX* resistance cassette (loxP-TEFp-KanMX-TEFt-loxP) from WSB118, all using PCR. Note that the *KanMX* resistance cassette contains 3' homology to the *ste3::HygMX* locus. Primers used contained 20 bp homology when appropriate to allow us to compile these sequences in the order listed using Gibson assembly. We then amplified this assembly further using PCR and transformed this 4.8 kB product into the ancestor (WY1335), screening for successful integration by G418 resistance, loss of hygromycin resistance, and checking PCR. We obtained WY2254 ~2255.

To knockout duplicated *LYP1* from *DISOMY14*, we amplified a 532 bp region upstream of *LYP1*, and a 739 bp region downstream of *LYP1*. We also amplified *KanMX* resistance. We assembled the three pieces via Gibson assembly (*LYP1* upstream, *KanMX*, *LYP1* downstream), PCR amplified the assembled molecule, and transformed *DISOMY14* cells with the PCR product. Transformants were plated on YPD + G418 plate, and colonies were screened for correct integration via PCR. We obtained WY2262 ~2263.

We used a similar methodology to introduce an extra copy of *WHI3* in the *ste3::HygMX* locus of the ancestor (WY1335). We first digested WSB185 (pBSKII) with XmaI and HindIII (HF), yielding 2.9 kb backbone. We amplified a 428 bp homology region upstream of *STE3*, the *WHI3* gene (including 570 bp upstream and 700 bp downstream of ORF;~3.3 kb), and *KanMX* resistance (1.6 kb), and assembled all these with the vector backbone. The Gibson product was used to transform *E. coli*, and colonies were mini-preped and screened via restriction digest with Kpn1 and BamH1. The correct liberated fragment (5.4 kb) was transformed into ancestral cells (WY1335) and plated on YPD + G418 plate. Transformants were checked for loss of hygromycin resistance and via PCR. We obtained WY2357 ~2359.

To delete *WHI3* from *DISOMY14*, we amplified a 528 bp region upstream of *WHI3*, and a 489 bp region downstream of *WHI3*. We assembled the two pieces with KanMX cassette via Gibson assembly, PCR amplified the assembled molecule, and transformed *DISOMY14* cells with the PCR product. Transformants were plated on YPD + G418 plate, and colonies were screened for correct integration via PCR. We obtained WY2350 ~2352.

## Genomic DNA extraction for sequencing

For whole-genome sequencing via tagmentation, we extracted yeast genomic DNA using QIAGEN Genomic-tip 20G (Cat. No. 10223), YeaStar Genomic DNA kit (Zymo Research), or a protocol modified from Sergey Kryazhimskiy and Andrew Murray lab. DNA from the last protocol is suitable for tagmentation but not for RADseq.

To extract DNA, we used the following procedure: "Spin down cells (0.5 ml saturated culture; microfuge at highest speed in 2 ml v-bottom tubes for 2'). Thoroughly discard supernatant. To the pellet add 252 µl Digestion mix (50 µl of 0.5M EDTA, pH = 7.5 or 8.0; 200 µl of ddH20; 2.5 µl of Zymolyase (stock: 5 U/ul)). Mix by inversion and incubate at 37°C for 24 hr on rotator. Add 50 µl of miniprep mix (0.2M EDTA, 0.4M Tris, 2% SDS, pH = 8.0). Mix by inversion and incubate at 65°C for 30 min. Add 63 µl (about ⅕ of volume) of 5M KAc. Mix by inversion and incubate on ice for 30 min. Add 250 µl of chloroform, vortex vigorously for 1 min. This helps to precipitate proteins and lipids. Spin down sample for 10 min on max speed (tabletop centrifuge). The DNA is in the supernatant. Transfer supernatant to a new tube (max 300 µl). Add 720 µl (>2 x volume) of 100% EtOH. Mix by inversion. At this point you should see the DNA clots in your tube. Spin down on max speed for 20 min. The DNA is now in the pellet. Thoroughly discard supernatant. Add 50 µl $H_2O$ +1 µl of RNAase A (10 mg/ml) to undried pellet. Allow DNA to resuspend by incubating at 37°C for 1 hr. Add 2 µl of Proteinase K (20 mg/ml). Incubate for 2 hr at 37°C. Add 130 µl of 100% isopropanol. Mix by inversion and spin down at max speed for 10 min. The DNA is in the pellet. Thoroughly discard supernatant. Add 500 µl of 70% EtOH. Mix by inversion and spin down at max speed for 10 min. The DNA is in the pellet. Discard supernatant. Allow pellet to air dry overnight. Resuspend pellet in 100 µl of 10 mM Tris, pH = 8.0."

For RADseq, yeast genomic DNA was extracted from $2 \times 10^8 \sim 10^9$ cells using, for example, the DNeasy Blood and Tissue Kit (Qiagen). High-quality DNA is required for optimal restriction

endonuclease digestion and is of utmost importance for the overall success of the protocol. The samples were treated with RNase A following manufacturer's instructions to remove residual RNA, and then quantified using Qubit. The optimal concentration after elution is 25 ng/µl or greater.

## Whole-genome sequencing

The whole genome sequencing protocol was slightly modified from that of Sergey Kryazhimskiy v. 2.1 (2013-06-06) (*Kryazhimskiy et al., 2014*). Indexing primer design followed (*Adey et al., 2010*) (*Supplementary file 4*). For an illustration of Nextera V2 Illumina sequencing molecular biology, see *Figure 2—figure supplement 2*.

To tagment genomic DNA, we prepared gDNA at concentration at or below 2.5 ng/µl. For $n$ samples = $r$ rows and $c$ columns, make the Tagmentation Master Mix (TMM) by mixing $n$ x 1.06 × 1.25 µl of TD Buffer (Tagment DNA Buffer) and $n$ x 1.06 × 0.25 µl of TDE1 (Tagment DNA Enzyme) in a PCR tube. Mix thoroughly by gently pipetting the mixture up and down 20 times. Distribute TMM into $r$ tubes (or a PCR strip), $c$ x 1.03 × 1.5 µl into each tube. With a multichannel pipette, distribute TMM into all wells of a fresh plate ('tagmentation plate'), 1.5 µl per well. With a multichannel pipette, transfer 1 µl of gDNA into the tagmentation plate (total volume = 2.5 µl per well). Mix by gently pipetting up and down 10 times. Cover plate with Microseal 'B' (Biorad, MSB-1001). Give the plate a quick spin to collect all liquid at the bottom (Sorvall or Allegra centrifuges, 1000 rpm for 1 min). Place the plate in the thermocycler and run the following program: 55°C for 5 min; hold at 10° C.

Next, PCR amplification is performed to add the index adaptors to tagmented DNA. Make the adaptor PCR reaction final volume to be 7.5 µl (2.5 µl of tagmented DNA from above, 3.75 µl of 2x KAPA master mix (KAPA amplification kit KK2611/KK2612), 0.625 µl of Index Adapter 1, 0.625 µl of Index Adapter 2). For convenience, we have pre-mixed index primers where index adapter one is always the same (NexV2ad1noBC; *Supplementary file 4*) and index adapter two is one of the 96 Index adaptors (NexV2ad2**; *Supplementary file 4*), and each is at 5 µM in H$_2$O. Mix the entire mix by gently pipetting up and down 10 times. Cover plate with Microseal 'A' (Biorad, MSB-5001). Make sure to press well on each well, especially edge wells. Give the plate a quick spin to collect all liquid at the bottom at 1000 rpm for 1 min. Place the tubes in the thermocycler and run the following program: 72°C for 3 min; 98°C for 2:45 min; [98°C for 15 s; 62°C for 30 s; 72°C for 1:30 min]x8; Hold at 4° C. Ensure that the lid is tight and that it is heated during incubation.

Make Reconditioning PCR Master Mix (RMM) by mixing $n$*1.06*8.5 µl of 2xKAPA polymerase mix, $n$*1.06*0.5 µl of primer P1 (10 µM; WSO380 AATGATACGGCGACCACCGA), and $n$*1.06*0.5 µl of primer P2 (10 µM; WSO381 CAAGCAGAAGACGGCATACGA). Mix thoroughly by gently pipetting the mixture up and down 20 times. With a multichannel pipette, transfer 9.5 µl of RMM into each well of the plate (final PCR volume 17 µl). Mix by gently pipetting up and down 10 times. Cover plate with Microseal 'A'. Give the plate a quick spin to collect all liquid at the bottom at 1000 rpm for 1 min. Place the tubes in the thermocycler and run the following program: 95°C for 5 min; [98°C for 20 s; 62°C for 20 s; 72°C for 30 s]x4; 72°C for 2 min; Hold at 4°C.

PCR clean-up used magnetic beads. Centrifuge the plate to collect all liquid (1000 rpm for 1 min). Vortex AMPure XP beads (Beckman Coulter A63880) for 30 s to ensure that they are evenly dispersed. Transfer $c$ x 1.05 × 1 x $V$ µl ($V$ = PCR volume = 17 µl) of beads into $r$ PCR tubes or a PCR strip. Using a multichannel pipette, transfer $V$ µl of beads into each well containing the PCR product. Mix well by gently pipetting up and down 20 times. The color of the mixture should appear homogeneous after mixing. Incubate at room temperature for 5 min so that DNA is captured by beads. Place the plate on the magnetic stand (Life Technologies, Cat. #123-31D) and incubate for about 1 min to separate beads from solution. Wait for the solution to become clear. While the plate is on the magnetic stand, aspirate clear solution from the plate and discard. Do not disturb the beads. If beads are accidentally pipetted, resuspend them back, wait for the solution to clear up, and repeat. While the plate is on the magnetic stand, dispense 200 µl of 70% ethanol into each well and incubate for 30 s at room temperature. Aspirate out ethanol without disturbing the beads and discard. Repeat for a total of 2 washes. Remove the remaining ethanol with P10 pipette. Let the plate air dry for approximately 5 min. Do not overdry the beads. Take the plate off the magnetic stand. Add 33 µl of 10 mM Tris-HCl (pH 8) to each well of the plate. Carefully resuspend the beads by mixing 10–15 times. Incubate for 2 min at room temperature. DNA is now in the solution. Place the plate back onto the magnetic stand and incubate for about 1 min to separate beads from solution. Wait for the

solution to become clear. While the plate is on the magnetic stand, aspirate clear solution from the plate and transfer to a fresh plate. Do not disturb the beads. If beads are accidentally pipetted, resuspend them back, wait for the solution to clear up, and repeat. Qubit quantify samples using 1 µl of eluate. We get about 6 ng/µl. Send 3 µl for High Sensitivity tapestation (75–1000 pg/µl) to get average length. If Qubit reading was >0.38 ng/µl, then sequencing would generally work.

Pool samples at equal molarity, with the final pool ideally being at least 2 nM (although 1 nM seemed fine as well). If we have 100 indexes, then each sample needs to be diluted to 0.02 nM. Qubit the pooled sample and submit 30 µl at 2 nM for sequencing on Illumina HiSeq 2000 (paired-end; 50 ~ 150 cycles; Nextera sequencing primers).

## RADseq

RADseq (restriction site-associated DNA sequencing) protocol was obtained from Aimee Dudley lab based on *Etter et al. (2012)*. The design scheme is in *Figure 3—figure supplement 4*, and primer sequences are in *Supplementary file 5*. Briefly, genomic DNA was digested with the six-cutter Mfe1 and the four-cutter Mbo1 (see below). Our desired DNA fragment would be flanked by Mfe1 and Mbo1 sites. To the digested DNA, we ligate annealed primers (P1 top annealed with P1 bottom containing a 4 bp barcode and Mfe1 overhang, and P2 top annealed with P2 bottom containing a 6 bp barcode and Mbo1 overhang). The dual barcode system allows many samples (e.g. 900 samples) to be sequenced simultaneously. P1 also contains Illumina Read One sequencing primer which will read 4 bp barcode and genomic DNA adjacent to the Mfe1 site, and P2 also contains Illumina Read Two sequencing primer and Index sequencing primer which will read 6 bp barcode and genomic DNA adjacent to the Mbo1 site.

The ligation product can be PCR amplified using WSO381 and WSO398 (*Figure 3—figure supplement 4*). P2 top primer has a stretch of sequences (lower case) that does not anneal with P2 bottom. Thus, in PCR round 1, only WSO398 is effective. In PCR round two or later, both WSO398 and WSO381 are effective. Note that the non-annealing sequence is designed for the following reason: Because most genomic DNA fragments are flanked on both sides by the 4-cutter Mbo1 site and these fragments would ligate with P2 on both sides, the non-annealing DNA segment (lower case) ensures that fragments lacking P1 cannot be amplified with WSO381.

To anneal P1 and P2, 100 µM stock plates of P1 (top and bottom) and P2 (top and bottom) primers were obtained from the Dudley Lab at the Pacific Northwest Research Institute. The P1 bottom primer and the P2 top primer include a 5'-phosphate modification required by ligation. For the P1 annealing reaction, the following was mixed: 10 µl 5M NaCl, 100 µl 1M Tris (pH 8.0), 888 µl $H_2O$, and 1 µl each 100 µM P1 top and P1 bottom primers (final concentration 100 nM). For the P2 annealing reaction, the following was mixed: 7 µl 5M NaCl, 70 µl 1M Tris (pH 8.0), 483 µl $H_2O$, and 70 µl each 100 µM P1 top and P1 bottom primers (final concentration 10 µM). The P1 and P2 top/bottom primer mixtures were aliquoted into PCR tubes at 100 µl per tube. The samples were heated at 95℃ for 1 min, then allowed to cool to 4℃ (at a rate of 0.1 ℃/sec) to allow annealing of the top and bottom primers. After cooling, the tubes were spun down and immediately put on ice. The P1 aliquots were consolidated into a single tube, and P2 aliquots were consolidated into a single tube.

The P1 and P2 annealed primers were combined as follows. 700 µl of P2 was diluted into 4.55 ml of $H_2O$ (diluted concentration at 1.33 µM). To 48 wells of a 96 well plate, 83 µl of this diluted annealed P2 was added to each well. 27.5 µl of the annealed P1 primers was added to each well. The final P1 +P2 vol is 110.5 µl (P2 final concentration 1 µM; P1 final concentration of 25 nM). It is very important to keep adapters cold after annealing. Specifically, keep at 4℃, mix on ice, thaw on ice after retrieving from −20℃ storage.

RNase-treated yeast genomic DNA was subjected to restriction digestion by Mfe1 and Mbo1 (NEB). Per reaction, 5.25 µl of the purified genomic DNA (~125 ng, Qubit quantified) was mixed with 0.625 µl 10X NEB Buffer 4 (or CutSmart buffer), 0.25 µl MboI (2.5 units), and 0.125 µl MfeI-HF (2.5 units) in PCR tubes (final volume of 6.25 ul). The mixture was set up on ice and incubated at 37℃ for 1 hr, and was heat inactivated by incubating at 65℃ for 20 min.

Custom adapters were annealed to the digested DNA using NEB T4 DNA ligase. Specifically, the following components were added to a tube in this order: 2.5 µl of the combined annealed P1 (25 nM)+P2 (1 µM) adaptor mix, the entire 6.25 µl digestion reaction, and 3.75 µl of a T4 ligase master mix (0.1 µl T4 ligase at 2000 units/µl, 1.25 µl T4 ligation buffer, and 2.4 µl $H_2O$). The reaction was combined on ice and mixed by pipetting up and down. Ligation was carried out at room

temperature for 20 min and heat inactivated in thermocycler at 65°C for 20 min. The samples were allowed to cool slowly to room temperature (30 min).

Every 24 samples were pooled together and concentrated using the QIAGEN MinElute PCR purification kit, and the MinElute column was eluted in 10 µl EB. The concentrated samples were subjected to gel electrophoresis using 2% low range ultra agarose gel (48 samples/lane). Sufficient band separation is achieved when the loading dye is approximately halfway down the gel, which ensures that the 150 band is separated from the 100 bp primer dimer. The size range 150 to 500 bp was excised out of the gel under a long-wavelength UV lamp and purified using the QIAGEN MinElute Gel Extraction Kit, eluting with 20 µl EB.

The gel extracted DNA was amplified using the NEBNext PCR Master Mix (NEB#M0541S) with custom primers (WSO398 and WSO381; *Supplementary file 5*). The mixture contained: 25 µl NEBNext PCR Master Mix; 1 µl WSO398 (10 µM); 1 µl WSO381 (10 µM); 1 µl DNA (5 ~ 10 ng), and 22 µl $H_2O$ (total 50 µl). PCR cycling was: 98°C (1 min); [98°C (10 s)+60°C (30 s)+72°C (30 s)]x14 cycles + 72°C (4 min)+4°C hold. The PCR reaction was cleaned and concentrated using the QIAquick PCR Purification Kit (QIAGEN), eluting in 30 µl $H_2O$. The expected concentration is ~30–40 ng/µl. The library quality (fragment size distribution) was ascertained using Tapestation. The resultant DNA was subjected to paired-end 25 cycles on Illumina HiSeq 2000 using TruSeq Dual Index Sequencing Primers.

## Sequence analysis

To analyze whole genome sequencing, a custom Perl script incorporating bwa (*Li and Durbin, 2009*) and SAMtools (*Li et al., 2009*) written by Robin Green was used to align paired-end reads to the *S. cerevisiae* RM-11 reference genome. Mutations were identified via GATK for single-nucleotide variants and indels, and cn.MOPs for local copy-number variant calling. A custom Perl script incorporating vcftools was used to automate comparison between ancestral versus evolved strains. All genetic changes were visually inspected using the Integrated Genome Viewer (IGV) environment for quality inspection and validation. Ploidy was calculated using custom python and R scripts wherein read depth was counted for each base. These read depths were averaged within successive 1000 bp windows; each window average is normalized by the median of all window averages across the genome. The normalized values for each window are log2 transformed and plotted versus the respective genomic position (chromosome/supercontig) for ease in graphical inspection of ploidy changes. Sequence analysis code can be publicly accessed at https://github.com/robingreen525/ShouLab_NGS_CloneSeq (*Green, 2019*; copy archived at https://github.com/elifesciences-publications/ShouLab_NGS_CloneSeq).

RADseq analysis was performed as described in *Etter et al. (2012)* using custom python and R scripts. Samples were split by their respective barcode and aligned to the RM11 reference genome using bwa. Up to six mismatches were allowed per read/marker. Next, reads with Phred quality scores below 20 or with a median coverage of less than two per sample were discarded. To ensure that each respective marker was representative of a properly digested MfeI-MboI, the expected length of each fragment based on a theoretical digestion of the RM11 genome was compared to the length of the actual marker as determined by read alignment to that marker (i.e finding reads that fell within the expected coordinates of a MfeI-MboI digest product). Next, for each marker, the proportion of reads aligning to that marker was normalized against total read alignment to the genome.

To ensure that only high quality markers were used, the CV of each marker across all tested strains were analyzed and markers with a CV of >= 0.6 were discarded. Additionally, only markers that were within the expected gel cut size of >125 bp and <400 bp were used. This still allowed >2000 markers to be used for downstream analysis.

To assess ploidy for RADseq, the same analysis was performed on a panel of 10 euploid strains. For each strain and for each marker, the relative proportion of that maker of the total reads for the strain of interest was compared against the median proportion of the total reads for the euploid panel. A supercontig (the RM11 assembly does not have full chromosomes but supercontigs) was called as duplicated if the average proportion of all makers on that supercontig in the backcrossed strain was 2-fold greater than the euploid panel. All disomy 14 calls for a tetrad segregated 2:2 as expected.

## Microcolony assay

This method has been described in *Hart et al. (2019b)*. Briefly, to assay for self-serving phenotype of an $L^-H^+$ mutant, we diluted a saturated overnight 1:6000 into SD +164 µM lysine, and allowed cultures to grow overnight at 30°C to exponential phase. We washed cells 3x with SD, starved them for 4–6 hr to deplete vacuolar lysine stores, and diluted each culture so that a 50 µl spot had several hundred cells. We spotted 50 µl on SD plate supplemented with 1.5 µM lysine (10 spots/plate), and allowed these plates to grow overnight. When observed under a 10x objective microscope, evolved cells with increased lysine affinity would grow into 'microcolonies' of ~20 ~ 100 cells, while the ancestral genotype would fail to grow (*Figure 2—figure supplement 1*). *DISOMY14* exhibited an intermediate phenotype where smaller microcolonies with variable sizes formed.

## Flow cytometry

Beads (ThermoFisher Cat R0300, 3 µm red fluorescent beads) were autoclaved in a factory-clean glass tube, diluted into sterile 0.9% NaCl, and supplemented with sterile-filtered Triton X-100 to a final 0.05% (to prevent bead clumping). The mixture was sonicated to eliminate bead clusters and was kept at 4°C in constant rotation to prevent settling and re-clumping. Bead density was quantified via hemacytometer and Coulter counting ($4-8 \times 10^6$ beads/ml final). The prepared bead mixture served as a density standard. Culture samples of interest were diluted to OD 0.01 ~ 0.1 ($7 \times 10^5$ - $7 \times 10^6$ cells/ml) in filtered water. Bead-cell mixtures were prepared by mixing 90 ul of the diluted culture sample, 10 µl of the bead stock, and 2 µl of 1 µM ToPro 3 (Molecular Probes T-3605), a nucleic acid dye that only permeates cell membranes of dead cells. Triplicate cell-bead mixtures were prepared for each culture in a 96-well format for high-throughput processing. Flow cytometry of the samples was performed on Cytek DxP Cytometer equipped with four lasers, ten detectors, and an autosampler. GFP ($H^-L^+$), mCherry ($L^-H^+$), and ToPro (dead cells) are respectively detected by 50 mW 488 nm laser with 505/10 (i.e., 500–515 nm) detector, 75 mW 561 nm Laser with 615/25 detector, and 25 mW 637 nm laser with 660/20 detector. Each sample was individually analyzed using FlowJo software to identify the number of beads, dead cells, and live fluorescent cells. Live and dead cell densities were calculated from the respective cell:bead ratios, corrected for the initial culture dilution factor. The mean cell density from triplicate measurements was used (coefficient of variation within 10%).

## Microscopy growth assay

See *Hart et al. (2019b)* for details on microscopy and experimental setup, method validation, and data analysis. Briefly, cells were diluted to low densities to minimize metabolite depletion during measurements. Dilutions were estimated from culture OD measurement to result in 1000 ~ 5000 cells inoculated in 300 µl SD medium supplemented with different metabolite concentrations in wells of a transparent flat-bottom microtiter plate (e.g. Costar 3370). We filled the outermost wells with water to reduce evaporation.

Microtiter plates were imaged periodically (every 0.5 ~ 2 hr) under a 10x objective in a Nikon Eclipse TE-2000U inverted fluorescence microscope. For each well, four adjacent positions were imaged. The microscope was connected to a cooled CCD camera for fluorescence and transmitted light imaging. The microscope was enclosed in a temperature-controlled chamber set to 30°C. The microscope was equipped with motorized stages to allow z-autofocusing and systematic xy-scanning of locations in microplate wells, as well as motorized switchable filter cubes capable of detecting a variety of fluorophores. Image acquisition was done with an in-house LabVIEW program, incorporating bright-field autofocusing (*Hart et al., 2019b*) and automatic exposure adjustment during fluorescence imaging to avoid saturation. Condensation on the plate lid sometimes interfered with autofocusing. Thus, we added a transparent 'lid warmer' on top of our plate lid (*Hart et al., 2019b*), and set it to be 0.5°C warmer than the plate bottom, which eliminated condensation. We used an ET DsRed filter cube (Exciter: ET545/30x, Emitter: ET620/60 m, Dichroic: T570LP) for mCherry-expressing strains.

Time-lapse images were analyzed using an ImageJ plugin Bioact (*Hart et al., 2019b*). Bioact measured the total fluorescence intensity of all cells in an image frame after subtracting the background fluorescence from the total fluorescence. A script plotted background-subtracted fluorescence intensity over time for each well to allow visual inspection. If the dynamics of four positions looked similar,

we randomly selected one to inspect. In rare occasions, all four positions were out-of-focus and were not used. In a small subset of experiments, a discontinuous jump in data appeared in all four positions for unknown reasons. We did not calculate rates across the jump. Occasionally, one or two positions deviated from the rest. This could be due to a number of reasons, including shift of focal plane, shift of field of view, black dust particles, or bright dust spots in the field of view. The outlier positions were excluded after inspecting the images for probable causes. If the dynamics of four positions differed due to cell growth heterogeneity at low concentrations of metabolites, all positions were retained.

We normalized total intensity against that at time zero for each position, and then averaged across positions. We calculated growth rate over three to four consecutive time points, and plotted the maximal net growth rate against metabolite concentration. If maximal growth rate occurred at the end of an experiment, then the experimental duration was too short and data were not used. For $L^-H^+$, the initial stage (3 ~ 4 hr) residual growth was excluded from analysis since residual growth was supported by vacuolar lysine storage.

## Bioassay

75 µl sample filtered through a 0.2 µm filter was mixed with an equal volume of a master mix containing 2xSD (to provide fresh medium) as well as tester cells auxotrophic for the metabolite of interest (~$1 \times 10^4$ cells/ml, WY1340 over-night culture) in a flat-bottom 96-well plate. We then wrapped the plate with parafilm and allowed cells to grow to saturation at 30℃ for 48 hr. We re-suspended cells using a Thermo Scientific Teleshake (setting #5 for ~1 min) and read culture turbidity using a BioTek Synergy MX plate reader. Within each assay, SD supplemented with various known concentrations of metabolite were used to establish a standard curve that related metabolite concentration to final turbidity (e.g. *Figure 3—figure supplement 5*). From this standard curve, the metabolite concentration of an unknown sample could be inferred.

## Release assay

Detailed description of the release assay during lysine starvation can be found in *Hart et al. (2019a)*. Briefly, $L^-H^+$ strain was pre-grown in synthetic minimal media SD supplemented with high lysine (164 µM) to exponential phase. The cultures were washed in lysine-free media and allowed to starve for 2 hr at 30℃ to deplete vacuolar lysine stores. Following starvation, the culture was periodically sampled (approximately every 6 hr for 24 hr) upon which live/dead cell densities were measured via flow cytometry (Materials and methods, 'Flow cytometry'), and culture samples were sterile filtered and supernatants were frozen. The supernatants were subjected to bioassay to measure hypoxanthine concentrations (Materials and methods, 'Bioassay'). Hypoxanthine release rate can be inferred by the slope of the linear function relating integrated live cell density over time (cells/ml*hr) versus measured hypoxanthine concentration (µM). For an example, see *Figure 3—figure supplement 1*.

To increase the throughput of this assay for screening chromosome truncation mutants (*Figure 3B*), we made the following modifications. After measuring time zero cell densities by flow cytometry, we loaded 200 µL of OD ~0.05 cells in SD per well and tracked fluorescence every 2 hr using automated 96-well plate fluorescence microscopy imaging (*Hart et al., 2019b*). The rest of each culture was treated as in the normal assay for sterile filtering at each sampling. Plate preparation, imaging, and images analysis were done as described in Materials and methods, 'Microscopy growth assay'. Fluorescence scales with live cell density (*Hart et al., 2019b*), so we were able to estimate live cell densities at each time point $t$ by taking (fluorescence intensity at time $t$)*(initial cell density)/(initial fluorescence intensity).

## Cell size measurements

Both Coulter counter and flow cytometry forward scattering can be used to compare the cell size distributions of yeast strains, with Coulter counter providing a direct measurement of cell size. We used the Z2 Coulter counter (Beckman), with the following settings: Gain = 128; Current = 0.5; Pre-amp Gain = 224. We diluted cultures to $OD_{600}$ ~ 0.01 to 0.3 (1 OD ~ $7 \times 10^7$ cells/ml) when necessary, sonicated cells (horn sonicator at low setting for three quick pulses or bath sonicator for 1 min), and

placed 100 µl culture into Coulter cuvette. We then added 10 ml sterilized isotone down the wall of the titled cuvette to avoid splashing, and analyzed the sample.

## Metabolite utilization in batch culture

We measured metabolite utilization after cells fully saturated the culture. We starved exponentially-growing cells (3–6 hr for $L^-H^+$, 24 hr for $H^-L^+$) to deplete initial intracellular stores and inoculated ~$1 \times 10^5$ cells/ml into various concentrations of the required metabolite up to 25 µM. We incubated for 48 hr and then measured cell densities by flow cytometry. We performed linear regression between input metabolite concentrations (horizontal axis) and final total cell densities (vertical axis) within the linear range, forcing the regression line through origin. Utilization per birth in a saturated culture was quantified from 1/slope.

## Calculating uncertainty of ratio

Since release rate and metabolite utilization were measured in independent experiments, their errors were uncorrelated. For ratio $f = A/B$, suppose that $A$ and $B$ have standard deviations of $\sigma_A$ and $\sigma_B$, respectively. Then $\sigma_f$ is calculated as $f\sqrt{(\sigma_A/A)^2 + (\sigma_B/B)^2}$.

## Chemostat

We have constructed an eight-vessel chemostat with a design modified from *Takahashi et al. (2015)*. For details of construction, modification, calibration, and operation, see *Skelding et al. (2017)*. A detailed discussion on using chemostats to quantify release and utilization phenotypes can be found in *Hart et al. (2019a)*. A summary is presented here.

For $L^-H^+$, due to rapid evolution, we devised experiments so that live and dead populations quickly reached steady state. We first calculated the expected steady state cell density by dividing the concentration of lysine in the reservoir (20 µM) by fmole lysine utilized per new cell (*Figure 4*). We washed exponentially growing cells to remove extracellular lysine and inoculated 50% ~ 75% of the vessel volume at 1/3 of the expected steady state density. We filled the rest of the 19 ml vessel with reservoir media (resulting in less than the full 20 µM of starting lysine, but more than enough for maximal initial growth rate, ~5–10 µM). We set the pump flow rate to achieve the desired doubling time $T$ (19 ml culture volume*ln(2)/$T$). We collected and weighed waste media for each individual culturing vessel to ensure that the flow rate was correct (i.e. total waste accumulated over time $t$ was equal to the expected flow rate*$t$). We sampled cultures periodically to track population dynamics using flow cytometry (Materials and methods, 'Flow cytometry'), and filtered supernatant through a 0.45 µm nitrocellulose filter and froze the supernatant for metabolite quantification at the conclusion of an experiment (Materials and methods, 'Bioassay'). At the conclusion of an experiment, we also tested input media for each individual culturing vessel to ensure sterility by plating a 300 µl aliquot on an YPD plate and checking for growth after two days of growth at 30°C. If a substantial number of colonies grew (>5 colonies), the input line was considered contaminated and data from that vessel was not used. For most experiments, we isolated colonies from end time point and checked percent evolved (Materials and methods, 'Microcolony assay'). For $L^-H^+$, we only analyzed time courses where >90% of population remained ancestral.

In a lysine-limited chemostat, live cell density $[L^-H^+]_{live}$ is increased by growth (at a rate $g$), and decreased by dilution (at a rate $dil$):

$$d[L^-H^+]_{live}/dt = (g - dil)[L^-H^+]_{live} \tag{S1}$$

$L$, lysine concentration in the culturing vessel, is increased by the supply of fresh medium (at concentration $L_0$), and decreased by dilution and utilization (with birth of each new cell utilizing $u_L$ amount of lysine).

$$dL/dt = L_0 \cdot dil - L \cdot dil - u_L g[L^-H^+]_{live} \tag{S2}$$

Finally, hypoxanthine concentration $H$ is increased by release (from live cells at $r_H$ per live cell per hr, *Hart et al., 2019a*), and decreased by dilution.

$$dH/dt = r_H \cdot [L^-H^+]_{live} - dil \cdot H \tag{S3}$$

Note that at the steady state (denoted by subscript 'ss'), growth rate is equal to dilution rate (setting *Equation S1* to zero):

$$g_{ss} = dil \tag{S4}$$

To measure metabolite utilized per birth at steady state, we set *Equation S2* to zero and also apply *Equation S4*

$$u_L = (L_0 \cdot dil - L_{ss} \cdot dil) / \left( g_{ss}[L^-H^+]_{live,ss} \right) \sim L_0 / [L^-H^+]_{live,ss} \tag{S5}$$

Here, the approximation holds because the concentration of lysine in chemostat ($L_{ss}$) is much smaller than that in reservoir ($L_0$) and thus $L_{ss}$ can be ignored.

To measure release rate at steady state, we can set *Equation S3* to zero and obtain:

$$r_H = dil \cdot H_{ss} / [L^-H^+]_{live,ss} \tag{S6}$$

Thus, the exchange ratio can be quantified from

$$\frac{r_H}{u_L} = \frac{dil \cdot H_{ss}/[L^-H^+]_{live,ss}}{L_0/[L^-H^+]_{live,ss}} = \frac{H_{ss}}{L_0} dil \tag{S7}$$

Again, $H_{ss}$ is the steady state hypoxanthine concentration in the chemostat culture vessel, $L_0$ is the lysine concentration in the reservoir, and *dil* is the dilution rate.

## Community growth rate

This derivation is adapted from *Hart et al. (2019a)*. If we culture $L^-H^+$ with $H^-L^+$, we have

$$\frac{d[L^-H^+]}{dt} = (b_L(L) - d_L - c_L)[L^-H^+] \tag{S8}$$

$$\frac{d[H^-L^+]}{dt} = (b_H(H) - d_H - c_H)[H^-L^+] \tag{S9}$$

$$\frac{dL}{dt} = r_L[H^-L^+] - u_L b_L(L)[L^-H^+] \tag{S10}$$

$$\frac{dH}{dt} = r_H[L^-H^+] - u_H b_H(H)[H^-L^+] \tag{S11}$$

*Equation S8* states that the clonal population density $L^-H^+$ increases at birth rate $b_L$ which in turn depends on the concentration of lysine $L$, and decreases at death rate $d_L$ and cost of metabolite overproduction $c_L$. *Equation S9* describes how clonal population density $H^-L^+$ changes over time. *Equation S10* states that the concentration of lysine $L$ increases due to releaser $H^-L^+$ releasing at a rate $r_L$ and decreases as $u_L$ amount is utilized per birth of consumer $L^-H^+$. *Equation S11* describes how the concentration of $H$ changes over time. All parameters are non-negative. Note that only a single genotype per species is considered.

We can calculate the steady state growth rate $g_{comm}$. Since strain ratio becomes fixed (*Figure 4—figure supplement 3*), both strains must grow at the same rate as the community. This also means that $L$ and $H$ concentrations do not change.

$$b_L - d_L - c_L = g_{comm}$$
$$b_H - d_H - c_H = g_{comm}$$
$$r_L[H^-L^+] = u_L b_L[L^-H^+] = u_L(g_{comm} + d_L + c_L)[L^-H^+]$$
$$r_H[L^-H^+] = u_H b_H[H^-L^+] = u_H(g_{comm} + d_H + c_H)[H^-L^+]$$

Multiply the last two equations, we get

$$r_H r_L = u_H u_L(g_{comm} + d_L + c_L)(g_{comm} + d_H + c_H)$$

Solving this, we get

$$g_{comm} = -\frac{(d_H + c_H + d_L + c_L)}{2} + \sqrt{\frac{r_H r_L}{u_H u_L} + \frac{(d_H + c_H - d_L - c_L)^2}{4}}.$$

When $d_H$ and $d_L$ and $c_H$ and $c_L$ are small compared to $\sqrt{\frac{r_H r_L}{u_L u_H}}$, which is the case for CoSMO, we have

$$g_{comm} \approx \sqrt{\frac{r_H r_L}{u_H u_L}}.$$

## Quantifying spatial growth dynamics

Briefly, exponentially growing $L^- H^+$ and $H^- L^+$ were washed free of lysine and hypoxanthine supplements, respectively. $H^- L^+$ cell were further starved for 24 hr to reduce CoSMO growth lag phase (*Hart et al., 2019a*). The two strains were then mixed at approximately 1:1 ratio, and 15 μl of $4 \times 10^4$ total cells were spotted on the center of agarose pads (1/6 of a petri dish pie), forming an inoculum spot of radius ~4 mm. Periodically, cells from pads were washed off into water and subjected to flow cytometry. The agarose pad generally contained 0.7 μM lysine, although including or not including this low concentration of lysine did not make a difference in the steady state community growth rate.

## Acknowledgements

We thank Aric Capel for chemostat experiments and sequencing results, Barbara Bengtsson for earlier work on CoSMO evolution, Gareth Cromie and Eric Jeffery in Aimee Dudley lab for advice on RADSeq and for providing reagents, Dan Gottschling for WSB174 (pADH4UCA-IV) plasmid, Sergey Kryazhimskiy for whole genome sequencing library preparation protocol, Klavins lab for Gibson assembly protocol, and Jacob Kitzman from Jay Shendure lab for bar-code index sequences. We thank Li Xie, Alex Yuan, and David Skelding for discussions, Maxine Linial and Delia Pinto-Santini for critical reading of the initial manuscript, and Kevin Foster (University of Oxford) and Ronald Noë for critical reading of the revised manuscript. This work benefited from the Fred Hutch High Performance Computing Service (funded by S10 SIG grant S10OD020069-01) and the Fred Hutch Genomics & Bioinformatics Shared Resource (funded by P30 Cancer Center Support Grant NCI 5 P30 CA015704).

## Additional information

### Competing interests

Wenying Shou: Reviewing editor, *eLife*. The other authors declare that no competing interests exist.

### Funding

| Funder | Grant reference number | Author |
| --- | --- | --- |
| National Institutes of Health | R01GM124128 | Samuel Frederick Mock Hart<br>Wenying Shou |
| National Institutes of Health | DP2 OD006498-01 | Samuel Frederick Mock Hart<br>Chi-Chun Chen<br>Wenying Shou |
| National Institutes of Health | 5R25HG007153-07 | Jose Mario Bello Pineda |
| W.M. Keck Foundation | Distinguished Young Scholars | Chi-Chun Chen<br>Wenying Shou |
| Fred Hutchinson Cancer Research Center | | Robin Green<br>Wenying Shou |
| National Science Foundation | Graduate Research Fellowship | Robin Green |

The funders had no role in study design, data collection and interpretation, or the decision to submit the work for publication.

## Author contributions
Samuel Frederick Mock Hart, Conceptualization, Data curation, Formal analysis, Investigation, Visualization, Methodology, Writing—original draft, Writing—review and editing; Jose Mario Bello Pineda, Data curation, Formal analysis, Investigation, Visualization, Methodology, Writing—original draft, Writing—review and editing; Chi-Chun Chen, Investigation, Writing—review and editing; Robin Green, Software, Visualization; Wenying Shou, Conceptualization, Resources, Data curation, Formal analysis, Supervision, Funding acquisition, Investigation, Visualization, Methodology, Writing—original draft, Project administration, Writing—review and editing

## Author ORCIDs
Samuel Frederick Mock Hart https://orcid.org/0000-0002-5068-2199
Jose Mario Bello Pineda https://orcid.org/0000-0003-1417-9200
Chi-Chun Chen https://orcid.org/0000-0002-5621-2008
Wenying Shou https://orcid.org/0000-0001-5693-381X

## Decision letter and Author response
Decision letter https://doi.org/10.7554/eLife.44812.032
Author response https://doi.org/10.7554/eLife.44812.033

# Additional files

## Supplementary files
• Supplementary file 1. Foresight, hindsight, insight, and the blur in-between.
DOI: https://doi.org/10.7554/eLife.44812.025

• Supplementary file 2. Strain table.
DOI: https://doi.org/10.7554/eLife.44812.026

• Supplementary file 3. Mutations in evolved $L^-H^+$.
DOI: https://doi.org/10.7554/eLife.44812.027

• Supplementary file 4. Whole genome sequencing primer sequences.
DOI: https://doi.org/10.7554/eLife.44812.028

• Supplementary file 5. RADseq primer sequences.
DOI: https://doi.org/10.7554/eLife.44812.029

• Transparent reporting form
DOI: https://doi.org/10.7554/eLife.44812.030

## Data availability
All data generated or analysed during this study are included in the manuscript and supporting files. Sequence analysis code can be publicly accessed at https://github.com/robingreen525/ShouLab_NGS_CloneSeq (copy archived at https://github.com/elifesciences-publications/ShouLab_NGS_CloneSeq).

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
