## [Decision Letter]

Thank you for submitting your article "Disentangling strictly self-serving mutations from win-win mutations in a mutualistic microbial community" for consideration by *eLife*. Your article has been reviewed by three peer reviewers, and the evaluation has been overseen by a Reviewing Editor and Diethard Tautz as the Senior Editor. The following individuals involved in review of your submission have agreed to reveal their identity: Antoine Frenoy (Reviewer #2).

The reviewers have discussed the reviews with one another and the Reviewing Editor has drafted this decision to help you prepare a revised submission.

In this study, Samuel Hart and coworkers investigate the evolution of an artificial mutualistic community consisting of two mutant yeast lines, one secreting lysine but needing hypoxanthine (L^+^H^-^), and the other relying on the secreted lysine while secreting hypoxanthine (L^-^H^+^). The authors find that mutants arise in the L^-^H^+^ line that alter the rate of lysine consumption and hypoxanthine secretion per cell. Specifically, duplication of a region of Chr14 caused duplication of the LYP1 lysine permease gene, resulting in increased lysine uptake. Similarly, duplication of the WHI3 gene results in larger cells that secrete more hypoxanthine per cell, with the secretion rate per amount of lysine remained unchanged, implying that while the mutated lines at first seem to also bestow a benefit for the L^+^H^-^ cells due to the increased hypoxanthine release, the net benefit of this mutation for the L^+^H^-^ cells is really zero, as the total number of mutant L^-^H^+^ cells is lower, resulting in an unchanged total hypoxanthine release.

Overall, this is rather simple, yet elegant study that shows a shortcoming of interpreting the effect of mutations on a mutualism when only considering one cell instead of the whole population. In light of these results, the authors propose a new model that, rather than calculating cost and benefit for each individual, takes into account the population-level cost and benefits.

Overall, we have two major & essential revisions:

1) We feel that a few simple yet essential experiments are lacking. The authors did not show experiments comparing the total growth/biomass production rate of the H^-^L^+^ interaction partner when growing with ancestral L^-^H^+^ versus the evolved disomy-14 L^-^H^+^ strains. The exchange ratio argument seems sound but in the end for a proper comparison to existing literature on social evolution experiments in microbes, the authors need to add and/or more clearly state what the total fitness effects are (to each strain alone and in co-culture) when the ancestral versus the disomy-14 L^-^H^+^ strains are grown with H^+^L-. In other words the theoretical framework in which the authors are defining cooperation versus antagonism or neutral interaction is not always consistent with the majority of literature in this area.

2) We feel that the results need to be put into a broader context and discussed in the light of existing literature. All three reviewers agreed that in its present form, the Discussion section of the paper is inadequate in that it provided little conceptual clarity on the scope or meaning of the results. Since some of the readership approaching this paper will be accustomed to sociobiology vernacular, we would encourage the authors to broaden their Discussion somewhat to include how their results fit into this framework. Discussion of the total fitness effects of disomy 14 in L^-^H^+^ on both interaction partners seems crucial in this respect (see major comment 1). Moreover, it would be useful to better compare this new model to existing ideas, e.g. the idea of calculating inclusive fitness, and critically discuss existing theory and previous studies to indicate to what extent the idea of calculating benefit/cost ratios at a population level is novel and generally applicable. We feel that the analogies with the fishes, the insects, and the biological market theory more confusing than enlightening, and they should be explained better. Lastly, the authors cite another recently accepted publication from their team (#25). As this publication is still "in press", it would be good to clearly compare the content of the accepted with the submitted publication and indicate what is really novel in the submitted publication.

---

## [Author Response]

1) We feel that a few simple yet essential experiments are lacking. The authors did not show experiments comparing the total growth/biomass production rate of the H-L+ interaction partner when growing with ancestral L^-^H^+^ versus the evolved disomy-14 L^-^H^+^ strains. The exchange ratio argument seems sound but in the end for a proper comparison to existing literature on social evolution experiments in microbes, the authors need to add and/or more clearly state what the total fitness effects are (to each strain alone and in co-culture) when the ancestral versus the disomy-14 L^-^H^+^ strains are grown with H^+^L-. In other words the theoretical framework in which the authors are defining cooperation versus antagonism or neutral interaction is not always consistent with the majority of literature in this area.

We have performed the experiment and obtained the expected result: H-L+ cocultured with ancestral L^-^H^+^ grew at indistinguishable rate as H-L+ cocultured with DISOMY14 L^-^H^+^. We have added to the main text:

“Consistent with *DISOMY14* not being partner-serving, partner *H^-^L^+^* grew at the same steady state rate whether co-cultured with ancestral or *DISOMY14 L^-^H^+^*(Figure 4E, right panel).”

“(E) *DISOMY14* and ancestral *L^-^H^+^* led to identical growth rate of community and of partner. To prevent rapid evolution in *L^-^H^+^*(Hart et al., 2019a), we grew CoSMO in a spatially-structured environment on agarose pads, and periodically measured the absolute abundance of the two strains (differentiable by their fluorescence; Methods, ‘Flow cytometry’). We then quantified the steady state growth rate of community and of partner *H^-^L^+^* by regressing ln(cell density) against time after the initial lag phase up to <10^8^ cells ((Hart et al., 2019a); see Fig 4 - figure supplement 3 for examples of detailed dynamics). We plotted the slope (i.e. growth rate), with error bars indicating 2x standard error of estimating the slope. In A, B, and E, we performed statistical comparisons first using the F-test to test for equal variance, and then using unpaired two-tailed t-test with equal variance. We plotted the corresponding P-values of the t-test (the probability of observing a test statistic as extreme as, or more extreme than, the observed value under the null hypothesis of two groups belonging to the same distribution). Comparisons in A and B show significant difference, while those in E are not significantly different. All data can be found in Figure 4—source data 1.”

2) We feel that the results need to be put into a broader context and discussed in the light of existing literature. All three reviewers agreed that in its present form, the Discussion section of the paper is inadequate in that it provided little conceptual clarity on the scope or meaning of the results. Since some of the readership approaching this paper will be accustomed to sociobiology vernacular, we would encourage the authors to broaden their Discussion somewhat to include how their results fit into this framework. Discussion of the total fitness effects of disomy 14 in L^-^H^+^ on both interaction partners seems crucial in this respect (see major comment 1). Moreover, it would be useful to better compare this new model to existing ideas, e.g. the idea of calculating inclusive fitness, and critically discuss existing theory and previous studies to indicate to what extent the idea of calculating benefit/cost ratios at a population level is novel and generally applicable. We feel that the analogies with the fishes, the insects, and the biological market theory more confusing than enlightening, and they should be explained better.

Thank you for this suggestion. I indeed wrote a “diet” version Discussions, partly because certain topics of social evolution are very contentious for reasons that are not entirely clear to some of its practitioners (myself included). In this revision, I hope that I have navigated some of these topics in a thoughtful fashion so that I would not provoke any antagonism. I did ask Prof. Kevin Foster (an expert in social evolution theory) and Prof. Ronald Noë (an expert in biological market theory) to double check my work, and I am very grateful to them for their help.

The Discussion has been extended to have three subsections ('Theories of mutualisms', 'Inclusive fitness', and 'Biological Market theory'), as it currently stands.

Lastly, the authors cite another recently accepted publication from their team. As this publication is still "in press", it would be good to clearly compare the content of the accepted with the submitted publication and indicate what is really novel in the submitted publication.

The article “Uncovering and resolving challenges of quantitative modeling in a simplified community of interacting cells” is now published https://journals.plos.org/plosbiology/article?id=10.1371/journal.pbio.3000135. The published article focused on how to properly measure phenotypes of the ancestral H^-^L^+^ and L^-^H^+^ strains so that we could quantitatively understand coculture growth rate. Because of this published article, we are confident about the phenotype measurements and the community growth measurements in this manuscript. The published article does not show data on *DISOMY14*, so the main messages do not overlap.